# TRIM: Token-wise Attention-Derived Saliency for Data-Efficient Instruction Tuning

**Manish Nagaraj** [1]  **Sakshi Choudhary** [1]  **Utkarsh Saxena** [1]  **Deepak Ravikumar** [1]  **Kaushik Roy** [1]

## Abstract

Instruction tuning is essential for aligning large language models (LLMs) to downstream tasks and commonly relies on large, diverse corpora. However, small, high-quality subsets, known as *coresets*, can deliver comparable or superior results, though curating them remains challenging. Existing methods often rely on coarse, sample-level signals like gradients, an approach that is computationally expensive and overlooks fine-grained features. To address this, we introduce TRIM (Token Relevance via Interpretable Multi-layer Attention), a forward-only, token-centric framework. Instead of using gradients, TRIM operates by matching underlying representational patterns identified via attention-based "fingerprints" from a handful of target samples. Such an approach makes TRIM highly efficient and uniquely sensitive to the structural features that define a task. Coresets selected by our method consistently outperform state-of-the-art baselines by up to 9% on downstream tasks and even surpass the performance of full-data fine-tuning in some settings. By avoiding expensive backward passes, TRIM achieves this at a fraction of the computational cost. These findings establish TRIM as a scalable and efficient alternative for building high-quality instruction-tuning datasets.

## 1. Introduction

Large language models (LLMs) have become the de facto standard for a wide range of tasks, but their full potential is unlocked only after a critical, computationally expensive step: instruction tuning, where a pretrained model is fine-tuned to align its outputs with user intent. While large, diverse instruction-tuning corpora have shown to be effective (Ouyang et al., 2022; Chung et al., 2024), a fundamental challenge persists: **not all data is created equal**. Recent findings have demonstrated that carefully curating a small, high-quality subset, or a "coreset", can not only match but even surpass the performance of finetuning on the full dataset, all while using a fraction of the compute (Xia et al., 2024; Zhang et al., 2025a).

Current solutions for obtaining coresets typically estimate sample-level importance using influence-style signals based on gradients or Hessians (Xia et al., 2024; Zhang et al., 2025b; San Joaquin et al., 2024). While effective, they are often costly to scale due to per-sample backward passes leading to large compute requirements (Pruthi et al., 2020). Cheaper alternatives based on representation similarity (Hanawa et al., 2021; Zhang et al., 2018) reduce compute by relying on forward-only embeddings, but still assign a single score per sample. It has been shown that sample-level importance estimation leads to two systematic biases: (i) **length bias**, since aggregating token-level signals (e.g., loss/gradient magnitudes) makes scores depend on sequence length, and gradient norms can be negatively correlated with length in instruction tuning (Xia et al., 2024); and (ii) **loss-centric bias**, since next-token prediction spreads credit uniformly across tokens even though attention and task-relevant evidence are typically sparse and localized (Zhang et al., 2023; Ge et al., 2023; Lin et al., 2024c; Jiang et al., 2024).

In principle, the most faithful measure of influence would come from token-wise gradients (or Hessians) (Lin et al., 2024a; Chen et al., 2026). However, this is infeasible at scale since it requires gradient/Hessian information at every token position in a sequence. We therefore aim to develop efficient proxies that preserve token-level resolution while avoiding the prohibitive cost of gradient computation. This leads to a central question: *"Given only a few target examples, how can we faithfully and efficiently identify high-impact instruction data by moving from sample-level scoring to token-level signals for task-specific fine-tuning?"*

To address this, we introduce **TRIM** (**T**oken **R**elevance via **I**nterpretable **M**ulti-layer Attention). [1] Our approach shifts

---

[1]Electrical and Computer Engineering, Purdue University. Correspondence to: Manish Nagaraj <mnagara@purdue.edu>.

*Proceedings of the 43rd International Conference on Machine Learning*, Seoul, South Korea. PMLR 306, 2026. Copyright 2026 by the author(s).

[1]Code available at https://github.com/manishnagaraj/TRIM_Tokenwise_Efficient_Finetuning

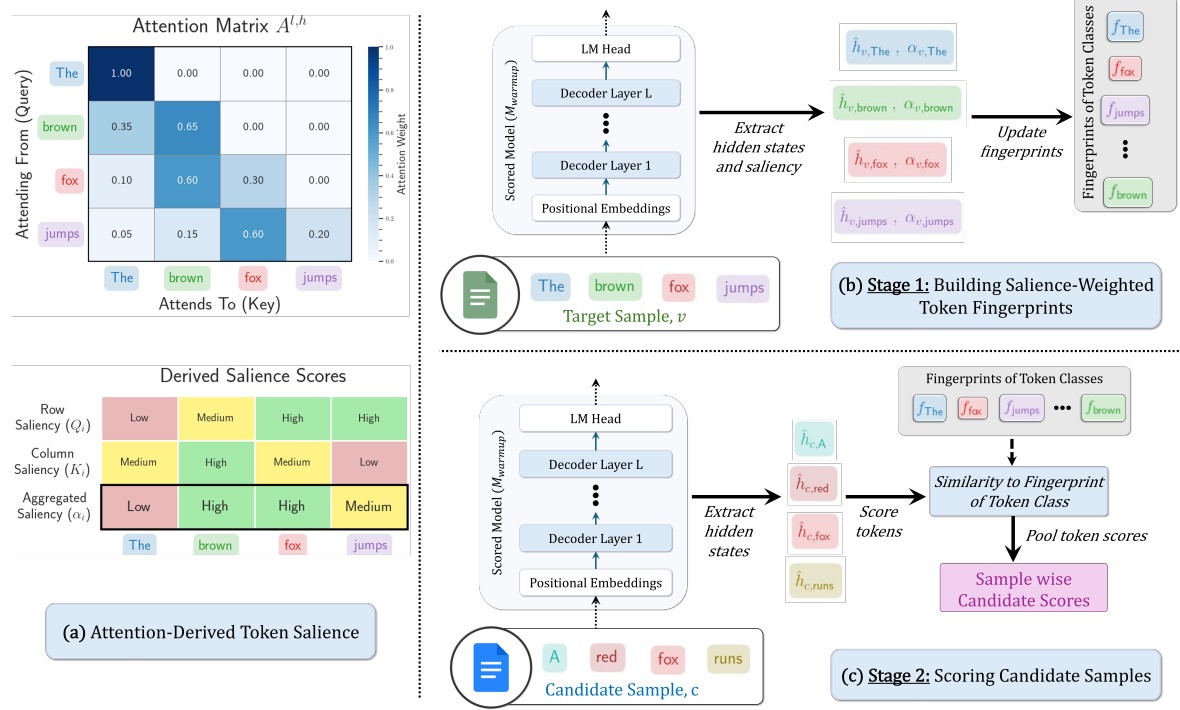

*Figure 1.* **Overview of** TRIM. (a) From multi-layer attention, we derive an aggregated token *saliency* signal that combines row (allocation sharpness) and column (received attention) signals. (b) Using a small target set, we compute one fingerprint per token class by saliency-weighted averaging of last-layer hidden states, capturing task-defining patterns (Section 3.1). (c) For each candidate sample, token states are matched to their class fingerprints (cosine similarity), and token scores are pooled into a single relevance score for ranking (Section 3.2); top-ranked samples form the instruction-tuning coreset.

from the traditional sample-level, loss-centric paradigm to a token-level, representation-centric one. The core insight is that a sample's utility often resides not in the entire sequence but in a few informative tokens (Jiang et al., 2024; Jin et al., 2024); thus, we leverage the model's attention layers to weight and match these critical representations to a target task.

As shown in Figure 1, TRIM is a two-stage pipeline requiring only forward passes. **Stage I** constructs lightweight token fingerprints from a small target set, representations that summarize how a given token class appears in the target context. It does so by computing an aggregated token saliency score that combines two complementary signals: row saliency, which measures how sharply a token distributes its own attention, and column saliency, which measures how much attention it receives from other tokens (panel a). For each token class, we then compute a fingerprint as the saliency-weighted mean of the last-layer hidden states of all target-set tokens in that class (panel b), yielding a concise summary of what that class looks like to the model. **Stage II** uses these fingerprints to score samples in the large candidate corpus. For each sample, we score token-level alignment to the task fingerprint (via cosine similarity; panel c) and pool these scores into a single relevance score.

We then select the top-scoring samples to form the final coreset. This design yields three advantages. First, TRIM delivers superior performance and efficiency: with only a handful of samples from the target task, it surpasses state-of-the-art methods (Xia et al., 2024; Zhang et al., 2025b; Yang et al., 2024) while running orders of magnitude faster, owing to its forward-only design that avoids per-sample gradient computation (Sections 4.1 and 5). Second, it offers high structural fidelity by matching core representational patterns (e.g., syntax, mathematical operators), which benefits challenging downstream tasks (Section 4.2). Finally, its token-level scoring inherently mitigates length bias, a common failure mode of sample-level methods (Section 4.4). Beyond targeted selection, we show that TRIM can be adapted to in-domain selection without a separate target set, using only the training pool itself (Section 4.5). Our contributions are:

- We introduce TRIM, a token-level, representation-based coreset selection method that uses attention-derived token saliency to score and select instruction-tuning examples from only a few target samples.

- TRIM is a forward-only algorithm that is orders of magnitude faster than gradient-based alternatives, enabling efficient data selection over massive instruction corpora.

- With only a 5% coreset and a handful of target samples, TRIM outperforms state-of-the-art methods by up to 9% on downstream reasoning tasks, and even exceeds full-data fine-tuning on some tasks.

## 2. Related Work

The challenge of curating coresets for instruction tuning has motivated a wide body of work on automated data selection (Cao et al., 2024; Chen et al., 2024a; Zhou et al., 2023; Ding et al., 2023). Existing methods can be grouped into several paradigms. **Influence-Based Selection** methods estimate data importance by quantifying the causal effect of a candidate example on a target task. As exact influence functions (Koh & Liang, 2017) are infeasible for LLMs, prior work relies on approximations, including Hessian-based variants (Kwon et al., 2024; San Joaquin et al., 2024; Lin et al., 2024b) and first-order alternatives such as check-pointed gradient similarity (e.g., LESS (Xia et al., 2024)). Forward-only proxies such as CLD (Nagaraj et al., 2025), DynUnc (He et al., 2024) correlate per-sample loss trajectories with validation dynamics, trading accuracy for efficiency. **Training-dynamics-based** methods identify informative examples without an explicit target gradient. Works such as TAGCOS and STAFF use gradient features from warmup checkpoints or surrogate models (Zhang et al., 2025b;c), while S2L transfers small-model loss signals to guide selection for larger models (Yang et al., 2024). In contrast, TRIM uses *targeted*, context-aware token-level selection from a few task samples. Recognizing the limitations of sample-level, **token-centric selection** uses token-level signals for pruning, long-context analysis, or filtering (Chen et al., 2025; Fu et al., 2026). Optimization-based approaches such as QCS (Chen et al., 2026) cast selection as a bi-level problem coupling sequence selection with token mining, which can be computationally heavy at scale. TRIM contributes to this direction by offering an efficient, reference-guided token-level selector that produces a single relevance score per example for scalable coreset construction. An extended discussion is provided in Appendix A.

## 3. Methodology

**Setup and Notation.** We begin with a pretrained language model $M_0$. Our goal is to select a compact coreset $\mathcal{C}$ from a large and diverse instruction-tuning corpus $\mathcal{S} = \{s_1, \ldots, s_{|\mathcal{S}|}\}$, which serves as the candidate pool for fine-tuning $M_0$ on a target domain $\mathcal{T}$. Coreset selection is guided by a small validation set $\mathcal{T}_{val}$ (disjoint from the test set), consisting of a very few samples from $\mathcal{T}$. Similar to prior works (Xia et al., 2024), we perform a brief *warmup* phase to adapt $M_0$ to the source distribution. Specifically, we fine-tune $M_0$ on a small random subset $\mathcal{S}_{warmup} \subset \mathcal{S}$ (about 5%), which produces the scoring model $M_{warmup}$.

All subsequent computations, i.e., attention-based saliency, fingerprint construction from $\mathcal{T}_{val}$, and candidate scoring over $\mathcal{S}$, are performed using only forward passes through $M_{warmup}$. TRIM then executes a forward-only, two-stage pipeline guided by $\mathcal{T}_{val}$. In Stage I (Section 3.1), it reads attention signals from $M_{warmup}$ to construct token-wise, task-defining fingerprints from the examples in $\mathcal{T}_{val}$. In Stage II (Section 3.2), each candidate $c \in \mathcal{S}$ is scored by comparing its token representations to the fingerprints, aggregating these comparisons into an example-level score, and selecting the top-ranked examples to form the final coreset $\mathcal{C}$.

### 3.1. Stage I: Building Saliency-Weighted Token Fingerprints

Given the target validation set $\mathcal{T}_{val}$, TRIM builds a compact dictionary of token fingerprints; prototype representations that capture how task-relevant tokens appear in context. This is achieved using only forward passes through $M_{warmup}$: we (i) compute a per-token importance weight from attention patterns, (ii) use these weights to aggregate contextual token representations from $\mathcal{T}_{val}$ into class-level fingerprints, and (iii) down-weight very common tokens (e.g., generic instruction formatting) using a corpus-frequency reweighting.

**Attention-Derived Token Saliency.** We define a token's saliency by integrating complementary signals from the row (query) and column (key) dimensions of the attention feature map (Figure 1a).

*Row Saliency.* We score a token at position $i$ by the row-wise entropy of its attention distribution. Tokens (query) with low row-wise entropy focus strongly on a few positions (keys) and are considered more salient, while those that spread attention broadly have low entropy and are less informative. For layer $l$, head $h$ and sequence length $T$, let $A^{l,h} \in \mathbb{R}^{T \times T}$ denote the attention matrix. We drop $h$ and $l$ notation for simplicity, and compute the entropy for $i^{th}$ row (query) as:

$$H_i = -\sum_j A_{i,j} \log(A_{i,j} + \varepsilon). \tag{1}$$

Here, $\varepsilon$ is added for numerical stability, and we normalize by the number of admissible keys under the attention mask (excluding padding; $j \leq i$ in a causal model) to obtain the row saliency score:

$$q_i = 1 - \frac{H_i}{\log |\{j : A_{i,j} > 0\}|}, \tag{2}$$

This is aggregated across the last $L$ layers and $H$ heads to produce a final row saliency score:

$$Q_i = \frac{1}{LH} \sum_{l=1}^{L} \sum_{h=1}^{H} q_i^{l,h} \in [0, 1]. \tag{3}$$

*Column Saliency.* The column saliency score is computed by aggregating all attention weights for each column (key) token. This signifies which tokens are attended strongly by others and are hence more likely to be informative. We quantify this by measuring average attention weights for $j^{th}$ column as follows:

$$k_j = \frac{\sum_i A_{i,j}}{|\{i : A_{i,j} > 0\}|} \tag{4}$$

Similar to row saliency, we aggregate this across heads and layers:

$$\overline{K}_j = \frac{1}{LH} \sum_{l=1}^{L} \sum_{h=1}^{H} k_j^{l,h} \tag{5}$$

Since $\overline{K}_j$ is unbounded, and its scale can vary across sequences, layers, and heads, we rescale it to $[0, 1]$ so it is comparable to $Q_i$ before combining the two signals. We therefore apply min–max normalization (with $\varepsilon$ added for numerical stability):

$$K_j = \frac{\overline{K}_j - \min \overline{K}_j}{\max \overline{K}_j - \min \overline{K}_j + \varepsilon} \in [0, 1]. \tag{6}$$

*Aggregated Token Saliency.* We perform a weighted averaging for the row and column saliency scores to obtain the aggregated saliency for the $i^{th}$ token:

$$\alpha_i = w_Q \, Q_i + w_K \, K_i. \tag{7}$$

This convex combination balances the complementary roles of row and column saliency, with higher $\alpha_i$ indicating greater token importance. For simplicity, we choose equal weights ($w_Q = w_K = 0.5$) for our experiments.

**Fingerprint Construction.** We now convert the saliency-weighted token signals into fingerprints: for each token class, we build a prototype vector that captures both how the token appears in the target context (via its hidden state) and how important that occurrence is (via its saliency). Fingerprints are defined at the token-class level, where each class corresponds to a unique token in the vocabulary (Wu & Papyan, 2024). Let $t$ denote a token class and let $h_{v,i}$ be the last-layer hidden state for an occurrence of $t$ at position $i$ in a target sample $v \in \mathcal{T}_{\text{val}}$. We collect all such occurrences:

$$O_t = \{(v,i) : v \in \mathcal{T}_{\text{val}}, \text{class}(v_i) = t\}. \tag{8}$$

We use last-layer hidden states since they are closest to the model's prediction head and thus provide a task-relevant token representation while keeping fingerprinting lightweight. The fingerprint for class $t$, denoted $f_t$, is then defined as the saliency-weighted average of the normalized hidden states $\hat{h}_{v,i} = \frac{h_{v,i}}{\|h_{v,i}\|_2}$:

$$f_t = \frac{\sum_{(v,i) \in O_t} \alpha_{v,i} \, \hat{h}_{v,i}}{\left\| \sum_{(v,i) \in O_t} \alpha_{v,i} \, \hat{h}_{v,i} \right\|_2}. \tag{9}$$

The resulting dictionary $\mathcal{F} = \{f_t\}$ compactly summarizes the validation set at the token level, emphasizing informative tokens through their saliency scores and contextual hidden representations.

**Frequency reweighting.** Eq. (9) weights token occurrences by attention-derived saliency $\alpha_{v,i}$, but very common tokens (e.g., generic instruction framing) can still dominate the aggregation simply because they appear frequently. To reduce this effect, we apply inverse document frequency (IDF) reweighting by replacing $\alpha_{v,i}$ with $\alpha_{v,i} \cdot \text{IDF}(t)$, where $t = \text{class}(v_i)$ and $\text{IDF}(t) = \log\left(\frac{N+1}{\text{df}(t)+1}\right)$ is computed over the candidate pool $\mathcal{S}$. We analyze the impact of IDF in the ablations (Appendix H).

### 3.2. Stage II: Candidate Scoring and Selection

With the aid of the fingerprint dictionary $\mathcal{F}$, TRIM assigns a score to each candidate sample $c \in \mathcal{S}$. To do so, we obtain the last-layer hidden states $h_{c,j}$ via a forward pass through $\mathbf{M}_{\text{warmup}}$. Intuitively, a candidate is valuable if its token representations align with the salient, task-defining fingerprints derived from $\mathcal{T}_{\text{val}}$. For each token at position $j$ in candidate $c$, let $t_j$ be its token class. We then compute its similarity score $s_j$ by measuring the cosine similarity between its normalized hidden state, $\hat{h}_{c,j} = \frac{h_{c,j}}{\|h_{c,j}\|_2}$, and the corresponding class fingerprint $f_{t_j}$.

$$s_j = \cos(\hat{h}_{c,j}, f_{t_j}). \tag{10}$$

**Handling Non-fingerprinted Token Classes.** If a token class $t_j$ does not appear in $\mathcal{T}_{\text{val}}$, it will not have a corresponding fingerprint in $\mathcal{F}$. For such tokens, we map $t_j$ to the closest fingerprinted class in the model's input-embedding space (by cosine similarity), and down-weight its contribution by a penalty factor $\lambda \in (0, 1]$. We precompute and cache this nearest-class mapping once per tokenizer/model (Appendix C). Let $E$ denote the input-embedding matrix of $\mathbf{M}_{\text{warmup}}$ and let $e_t$ be the unit-normalized embedding for class $t$. The fallback mapping is:

$$\bar{t}_j = \arg\max_{t \in \mathcal{F}} \cos(e_{t_j}, e_t), \tag{11}$$

and the penalized similarity score is:

$$s_j = \lambda \cdot \cos(\hat{h}_{c,j}, f_{\bar{t}_j}). \tag{12}$$

**Robust Sample Score.** The final score for a candidate $c$ is aggregated over its scored token set $\mathcal{J}(c)$ using a robust pooling function that combines the mean and maximum of

its token scores, together with a light coverage term:

$$\mu(c) = \frac{1}{|\mathcal{J}(c)|} \sum_{j \in \mathcal{J}(c)} s_j, \tag{13}$$

$$\tau(c) = \max_{j \in \mathcal{J}(c)} s_j, \tag{14}$$

$$\kappa(c) = \frac{|\mathcal{J}(c)|}{|c|}, \tag{15}$$

$$S(c) = w_\mu \mu(c) + w_m \tau(c) + \eta \kappa(c). \tag{16}$$

The mean term captures overall informativeness across tokens, while the max term highlights the presence of highly crucial tokens. The coverage term lightly rewards candidates for which scored tokens are less sparse within the sequence and helps mitigate length effects. In our experiments, we fix $w_\mu = w_m = 0.5$ and use a small $\eta$ so that coverage acts primarily as a tie-breaker rather than dominating the similarity-based score.

**Coreset Selection.** Finally, we rank all candidate examples $c \in \mathcal{S}$ by their scores $S(c)$ and select the top-ranking examples to form the coreset $\mathcal{C}$. Since this scoring stage only requires a single forward pass per candidate, the runtime is linear in the corpus size once the fingerprints are constructed. For detailed pseudocode, please refer to Appendix B.

# 4. Experiments

We evaluate TRIM along four axes: (i) budgeted accuracy on general-domain reasoning benchmarks and a challenging downstream math task (Sections 4.1 and 4.2); (ii) cross-model/scale transfer (Section 4.3); (iii) robustness to length bias (Section 4.4); and (iv) an in-domain setting to study coreset selection when there are no external target examples, with the subset selection relying on the candidate pool $\mathcal{S}$ itself (Section 4.5). Additional hyperparameter ablations, experiments on sensitivity to target samples, as well as performance on additional models and benchmarks, are provided in the Appendices G, H, I, J, and N.

**Experimental Setup.** For all experiments, we follow the warmup and selection protocol of Xia et al. (2024). We first fine-tune a scorer on a random 5% subset of the candidate pool $\mathcal{S}$ using LoRA (Hu et al., 2022), saving intermediate checkpoints. We then score all candidates in $\mathcal{S}$ using this warmed-up scorer model and select a budgeted coreset. Finally, we fine-tune a fresh target model initialized from the pretrained checkpoint on the selected coreset and report downstream performance.

**General instruction-pool setting.** Unless stated otherwise, we select from a large candidate pool of 270k *instruction-tuning* examples (DOLLY (Conover et al., 2023), OASST1 (Köpf et al., 2023), FLAN_V2 (Longpre et al.,

2023), and CoT (Wei et al., 2022)). Our target tasks are MMLU (Hendrycks et al., 2020), BBH (Suzgun et al., 2023), TYDIQA (Clark et al., 2020), and GSM8K (Cobbe et al., 2021). To unify heterogeneous sources, we format all datasets into a chat-style schema following the instruction-tuning pipeline of Wang et al. (2023). We evaluate with `lm-evaluation-harness` (Gao et al., 2023) using fixed few-shot prompting: 5-shot MMLU, 3-shot BBH, 1-shot TYDIQA, and 8-shot GSM8K with CoT. Additional dataset and evaluation details are in Appendices E and F.

**Models.** We use LLAMA-2-7B (Touvron et al., 2023) as the base model for all main experiments. To study transferability, we test whether coresets selected using a LLAMA-2-7B scorer transfer to LLAMA-2-13B (Touvron et al., 2023) and MISTRAL-7B (v0.3) (Jiang et al., 2023). Data selection is guided by a small target validation set per benchmark, where the number of validation examples is determined by the benchmark's few-shot configuration following Xia et al. (2024). The training details are provided in the Appendix G.

**Baselines.** We group coreset methods by mechanism and defer details to Appendix D. (i) Heuristics: Random, lexical retrieval (BM25 (Robertson et al., 2009)), and $N$-gram importance resampling (DSIR (Xie et al., 2023)). (ii) Forward-only training dynamics: CLD (correlation of loss differences) (Nagaraj et al., 2025) and S2L (loss-trajectory clustering) (Yang et al., 2024). (iii) Representation-based: RDS (Hanawa et al., 2021). (iv) Gradient-based: LESS (Xia et al., 2024) and TAGCOS (Zhang et al., 2025b). TRIM/TAGCOS score candidates using the final warm-start checkpoint, LESS/RDS aggregate scores across checkpoints, and CLD/S2L rely on trajectories collected during full-dataset training.

### 4.1. Performance Comparison

Table 1 compares the performance of various data selection methods at a fixed 5% budget. TRIM achieves the best overall performance, with a mean accuracy of **48.56%**, showing comparative performance to the strongest baseline, LESS (48.27%), while requiring substantially less computation. Notably, TRIM and LESS are the only methods whose 5% coresets surpass full-data fine-tuning on TYDIQA.

Beyond the 5% budget, Figure 2 shows TRIM consistently matching or improving upon strong baselines (LESS, TAGCOS) across coreset sizes. On MMLU and TYDIQA, TRIM remains competitive across all budgets and exceeds the full-data reference once moderate budgets are available; on BBH, it maintains a stable advantage over TAGCOS and tracks LESS closely as the budget increases. Overall, heuristic lexical methods (BM25, DSIR) primarily track surface overlap, while training-dynamics (CLD, S2L), representation-based (RDS), and gradient-based (LESS,

*Table 1.* Accuracy (%) for coreset methods on LLAMA-2-7B with a 5% coreset. Results are means over 3 seeds (standard deviations in parentheses). The final column is the macro-average across tasks.

| Coreset Method | MMLU | BBH | TYDIQA | AVERAGE MEAN |
|---|---|---|---|---|
| *Pretrained (no Fine-tuning)* | 45.60 | 38.30 | 46.40 | 43.43 |
| *Full-data Fine-tuning* | 50.12 (0.1) | 42.84 (0.1) | 54.0 (0.1) | 48.99 |
| Random | 45.84 (0.4) | 37.13 (0.4) | 52.45 (0.6) | 45.14 |
| BM25 (Robertson et al., 2009) | 46.12 (0.0) | 37.4 (0.0) | 52.70 (0.0) | 45.41 |
| DSIR (Xie et al., 2023) | 45.73 (0.1) | 34.70 (0.3) | 45.62 (0.5) | 42.02 |
| S2L (Yang et al., 2024) | 46.70 (0.2) | 37.24 (0.4) | 52.74 (0.3) | 45.56 |
| RDS (Hanawa et al., 2021) | 45.27 (0.6) | 35.13 (0.5) | 46.10 (0.2) | 42.17 |
| CLD (Nagaraj et al., 2025) | 46.13 (0.2) | 36.12 (0.3) | 45.58 (0.5) | 42.61 |
| LESS (Xia et al., 2024) | 49.23 (0.3) | 39.45 (0.2) | 56.13 (0.4) | 48.27 |
| TAGCOS (Zhang et al., 2025b) | 48.12 (0.2) | 38.24 (0.3) | 53.45 (0.4) | 46.60 |
| TRIM (**Ours**) | **49.33** (0.4) | **39.73** (0.89) | **56.62** (0.1) | **48.56** |

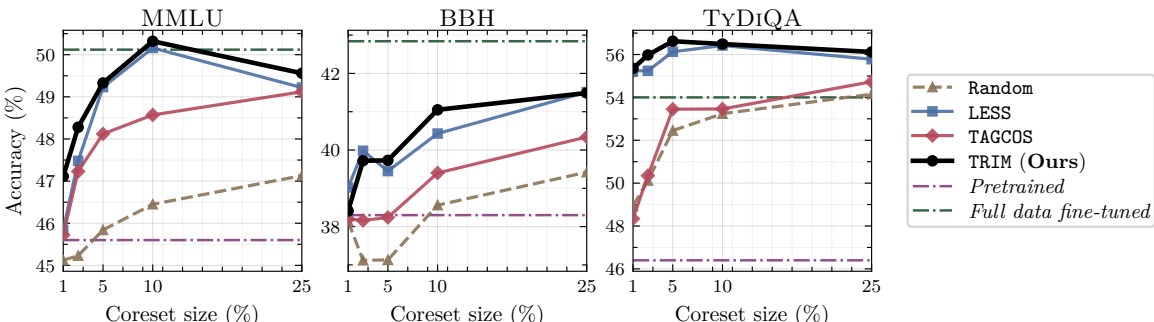

*Figure 2.* TRIM vs. top baselines on LLAMA-2-7B across coreset budgets for MMLU, BBH, and TYDIQA. TRIM maintains a consistent advantage and, on TYDIQA, exceeds the full-data baseline with coresets as small as 1%.

TAGCOS) selectors operate at the sample level and are therefore coarser. In contrast, TRIM's token-centric scoring emphasizes task-relevant tokens, enabling a precise selection.

### 4.2. Challenging Downstream Tasks

We rigorously evaluate the role of the token-centric approach in challenging downstream-task settings, where the target task has a low topical overlap with the candidate pool. Concretely, we ask each method to select a coreset for GSM8K from the same general-purpose instruction-tuning pool (not curated for math), then fine-tune on the selected data and evaluate on GSM8K. This setting is challenging: fine-tuning on the full pool yields only 30.25%, so success requires identifying examples with latent structure (e.g., stepwise logic, numerical manipulation) rather than superficial topical cues. As shown in Table 2, a 5% TRIM coreset reaches **29.52%**, nearly matching full-data fine-tuning and outperforming the next best method (LESS) by ~8.8%. Even at a 1% budget, TRIM achieves 22.33% accuracy, remaining well ahead of alternatives. This stems

*Table 2.* Challenging downstream data selection for GSM8K on LLAMA-2-7B. The table shows exact-match 8-shot accuracy (%) for coresets of 1% and 5% selected from a non-math corpus. Results are means over 3 seeds (standard deviations in parentheses).

| Coreset Method | $p = 1\%$ | $p = 5\%$ |
|---|---|---|
| *Pretrained (no Fine-tuning)* | 14.00 | |
| *Full-data Fine-tuning* | 30.25 (0.5) | |
| Random | 15.43 (0.80) | 18.21 (0.6) |
| S2L | 16.48 (0.4) | 18.55 (0.3) |
| LESS | 17.34 (1.2) | 20.72 (0.4) |
| TAGCOS | 17.23 (0.2) | 18.72 (0.4) |
| TRIM (**Ours**) | **22.33** (0.4) | **29.52** (0.3) |

from strong structural fidelity: by matching token-level fingerprints rather than sample-level losses or gradients, TRIM retrieves examples that convey a style of reasoning (e.g., enumerations, logical connectives, stepwise instruc-

tions). These signals act as transferable proxies for chain-of-thought structure in GSM8K, yielding gains that sample-level metrics fail to capture.

### 4.3. Coreset Transferability

We assess whether a scorer model can curate a coreset that transfers across scale and architecture. Using LLAMA-2-7B as the scorer, we select a single 5% TRIM coreset and reuse it to fine-tune two target models: LLAMA-2-13B (larger model within the same family) and MISTRAL-7B (V0.3) (a different architecture). We compare (i) TRIM-Transfer, where selection is performed once by the 7B scorer, against (ii) TRIM-Oracle, where the target model selects its own coreset (an in-model reference) in Table 3. On LLAMA-2-13B, TRIM-Transfer matches or exceeds the in-model oracle on MMLU and substantially improves TYDIQA, while remaining competitive on BBH (49.83 vs. 50.02). On MISTRAL-7B, TRIM-Transfer remains robust across all three benchmarks, outperforming full-data fine-tuning and approaching the oracle (TYDIQA: 60.43 vs. 59.83), though the oracle is strongest on MMLU and BBH. Overall, these results suggest that TRIM's token-level fingerprints capture transferable, model-agnostic structure, enabling a single scorer to select high-quality coresets that generalize across scale and architecture.

### 4.4. Mitigating Length Bias

A common failure mode in coreset selection is a bias toward shorter sequences. Sample-level scores can be confounded by length; for example, influence estimates derived from gradient magnitudes may favor short examples with concentrated signals over longer prompts containing richer reasoning (Xia et al., 2024). We analyze this effect by comparing the length distributions of coresets selected for GSM8K by TRIM, LESS, and TAGCOS. As shown in Figure 3, LESS and TAGCOS skew short (mean lengths 323 and 259 tokens). In contrast, TRIM selects a broader distribution with more long, complex examples (mean length 446 tokens), ∼38% longer than LESS and ∼72% longer than TAGCOS. This behavior is consistent with TRIM's token-centric scoring: relevance is assigned to salient tokens, and pooling aggregates token matches without directly rewarding shorter sequences.

### 4.5. In-Domain Data Selection Without Target Examples

So far, our selection setups assume access to a separate target/validation set (from the downstream task) to instantiate $\mathcal{T}_{\text{val}}$ for validation and fingerprinting. Here, we consider a stricter in-domain regime where such target examples are unavailable: selection must use only the candidate pool, aiming to preserve the pool distribution.

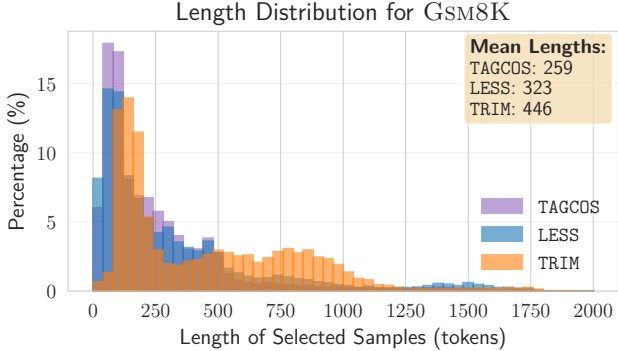

*Figure 3.* Length distribution of selected coresets for GSM8K. The histogram shows the percentage of selected samples by length bucket. Sample-level methods (LESS, TAGCOS) skew short, whereas TRIM selects a distribution with a higher mean length.

We instantiate this setting with the math-focused MATHIN-STRUCT corpus (Yue et al., 2024) as the candidate pool $\mathcal{S}$, formatted in the Alpaca chat schema (Chen et al., 2024b), and construct an in-domain reference set $\mathcal{T}_{\text{val}}$ by sampling $m{=}20$ examples uniformly from each of the 13 MATHIN-STRUCT sources ($|\mathcal{T}_{\text{val}}|{=}260$). Unlike prior sections where $\mathcal{T}_{\text{val}}$ is disjoint from $\mathcal{S}$, here $\mathcal{T}_{\text{val}} \subset \mathcal{S}$ by construction. We then follow the standard protocol, a warmed-up LLAMA-2-7B (trained on 5% random MATHINSTRUCT) scores candidates in $\mathcal{S}$; we select a 50% coreset; and fine-tune a fresh pretrained LLAMA-2-7B on the selected subset. Table 4 reports 0-shot exact-match on GSM8K and SVAMP under the MAMMOTH pipeline. TRIM matches prior works such as S2L/LESS[2] within $< 1\%$ while being substantially more compute-efficient (see Section 5). Additional details, ablations, and extended results are in Appendix M.

## 5. Computational Cost of Data Selection

We analyze the cost of the selection stage (model preparation + candidate scoring) and complement the asymptotic analysis with measured wall-clock runtimes. Let $N$ be the number of candidate examples and $T$ the number of training epochs used for model preparation (i.e., warmup). Let $f$ denote the FLOPs of a single forward pass of the scoring model. Following convention, a backward pass costs $\approx 2f$, so a training step (forward + backward) costs $\approx 3f$. We write $\gamma \in (0, 1]$ for the fraction of the candidate pool used during warmup and $C$ for the number of checkpoints whose states/representations are used during scoring. For a fair comparison, we assume all methods use a scoring model with forward-pass cost $f$. Finally, processing the target validation set is negligible compared to scoring the candidate pool since $N \gg Q$ (in our setup, $N{>}270$k while $Q \leq 285$).

With a fixed scoring model size, the main drivers are (i)

---

[2]To ensure fairness, $\mathcal{T}_{\text{val}}$ is the same for both TRIM and LESS.

*Table 3.* **Cross-Model and Cross-Scale Transferability of Coresets Selected by a 7B Scorer.** Results are means over 3 seeds (standard deviations in parentheses). **Bold** numbers denote the best performing selected subset. Underlined numbers denote that the selected subset outperformed the full-data fine-tuning.

| Coreset Method | LLAMA-2-13B | | | MISTRAL-7B (V0.3) | | |
|---|---|---|---|---|---|---|
| | MMLU | BBH | TYDIQA | MMLU | BBH | TYDIQA |
| *Full-data fine-tuning* | 54.14 *(0.0)* | 49.57 *(0.1)* | 53.87 *(0.0)* | 60.12 *(0.1)* | 53.0 *(0.0)* | 57.62 *(0.0)* |
| Random | 53.16 *(0.4)* | 46.48 *(0.2)* | 51.81 *(0.4)* | 60.10 *(0.3)* | 53.12 *(0.4)* | 55.18 *(0.4)* |
| TRIM-Transfer (from 7B) | **54.50** *(0.2)* | 49.83 *(0.6)* | **56.70** *(0.3)* | 60.40 *(0.8)* | 55.50 *(0.2)* | **60.43** *(0.4)* |
| TRIM-Oracle (in-model) | 54.10 *(0.6)* | **50.02** *(0.7)* | 54.12 *(0.2)* | **61.72** *(0.3)* | **56.12** *(0.4)* | 59.83 *(0.2)* |

*Table 4.* Exact-match 0-shot accuracy (%) on GSM8K and SVAMP for coreset methods on LLAMA-2-7B fine-tuned on a 50% coreset selected from MATHINSTRUCT. TRIM achieves accuracy comparable to prior in-domain selectors while using substantially lower selection compute (see Section 5). Results are means over 3 seeds (standard deviations in parentheses).

| Coreset Method | GSM8K | SVAMP |
|---|---|---|
| *Pretrained (no Fine-tuning)* | 3.10 | 12.40 |
| *Full-data Fine-tuning* | 51.15 *(0.2)* | 66.12 *(0.1)* |
| Random | 48.16 *(0.5)* | 63.30 *(1.2)* |
| S2L | 52.45 *(0.1)* | 65.30 *(0.2)* |
| LESS | 52.10 *(0.1)* | 65.45 *(0.2)* |
| TAGCOS | 50.20 *(0.2)* | 64.15 *(0.4)* |
| TRIM **(Ours)** | 52.23 *(0.2)* | 65.12 *(0.4)* |

*Table 5.* Selection cost and measured runtime (5% coreset). Theoretical cost is for the selection stage (warmup + scoring) under a common scoring model with forward-pass cost $f$. The runtime column reports the total wall-clock time (hours).

| Coreset Method | Asymptotic Selection Cost | Total Runtime (in Hours) |
|---|---|---|
| S2L | $\mathcal{O}(3fNT + fNC)$ | 87.95 |
| RDS | $\mathcal{O}(3f\gamma NT + fNC)$ | 20.48 |
| LESS | $\mathcal{O}(3f\gamma NT + 3fNC)$ | 35.76 |
| TAGCOS | $\mathcal{O}(3f\gamma NT + 3fN)$ | 14.16 |
| TRIM **(Ours)** | $\mathcal{O}(3f\gamma NT + fN)$ | **4.40** |

FLAN_V2), and a LLAMA-2-7B scorer selecting a 5% coreset for MMLU. All methods use the same warmup recipe, yielding a warmup time of 1.41 hours, *except* S2L, which requires full-data training to collect loss trajectories and therefore incurs a substantially larger preparation cost (27.96 hours). Table 5 reports the asymptotic selection cost alongside the measured total runtime. Overall, the measured runtimes mirror the asymptotic analysis: TRIM scores candidates using only forward passes (and a single checkpoint), making it considerably faster than methods that rely on gradients during scoring (LESS, TAGCOS) or full-data trajectory collection (S2L).

## 6. Conclusion

We introduced TRIM, a token-centric framework for efficient coreset selection in instruction tuning. By shifting from coarse, sample-level signals to fine-grained, attention-derived token representations, TRIM addresses two persistent challenges in data selection: computational cost and length bias. Through *saliency-weighted* fingerprints constructed from a handful of target samples, TRIM identifies training data that matches the structural patterns defining a task, rather than relying on expensive gradient computation or surface-level similarity. Our experiments demonstrate that TRIM consistently outperforms state-of-the-art meth-

whether scoring requires backward passes and (ii) how many checkpoints $C$ are used. LESS is most expensive at scoring time: it aggregates gradient features across multiple checkpoints, incurring a backward pass per candidate per checkpoint and yielding a $\mathcal{O}(3fNC)$ scoring term in addition to warmup $\mathcal{O}(3f\gamma NT)$. Forward-only methods that avoid gradients at scoring time (e.g., S2L, RDS) remove the $3f$ factor during scoring but still scale linearly with $C$ via $\mathcal{O}(fNC)$. Notably, S2L must train on *all* $N$ candidates to record per-sample loss trajectories (the signal it clusters), so its preparation term is $\mathcal{O}(3fNT)$ rather than $\mathcal{O}(3f\gamma NT)$. TAGCOS scores at a single checkpoint ($C{=}1$) but requires per-candidate gradients once, giving $\mathcal{O}(3fN)$ after warmup. In contrast, TRIM scores each candidate with a single forward pass at one checkpoint ($C{=}1$), producing $\mathcal{O}(fN)$ after warmup. Consequently, TRIM is asymptotically the most efficient among the compared methods for large $N$, and its benefits grow when baselines require multiple checkpoints ($C \gg 1$) or gradient computation during scoring.

To ground the above analysis, we measure end-to-end selection time (warmup + scoring) under a shared setup: a single NVIDIA H200 GPU, a common candidate pool of 270k instruction-tuning examples (from DOLLY, CoT, OASST1,

ods across several downstream tasks, achieving up to 9% improvements with only 5% coresets. Notably, TRIM's token-level scoring enables it to surface structurally relevant data even in low-overlap settings, as evidenced by near-full-data performance on mathematical reasoning using a general-purpose corpus. We also show that TRIM extends to setting where there are no separate target examples available, enabling effective curation directly from a specialized training pool. The method's forward-only design delivers these gains at a fraction of the computational cost of gradient-based alternatives, and coresets selected by small scorers transfer effectively to larger models across architectures. TRIM's ability to capture task-defining structure through attention signals offers a scalable path forward, enabling practitioners to curate smaller, higher-quality datasets.

## Acknowledgements

This work was supported in part by the Center for the Co-Design of Cognitive Systems (CoCoSYS), a research center under the Joint University Microelectronics Program (JUMP) 2.0, a Semiconductor Research Corporation (SRC) initiative sponsored by DARPA, Office of Basic Energy Sciences, and by the National Science Foundation.

## Impact Statement

TRIM improves the computational efficiency of instruction-tuning by enabling targeted coreset selection using forward-only signals and token-level structure matching. By reducing the amount of training data (and associated optimization steps) needed to achieve strong downstream performance, TRIM can lower the energy and cost of adapting language models, potentially broadening access to model customization for academic groups and smaller organizations. At the same time, TRIM is a data curation method: it changes *which* examples are used for training. If the candidate pool contains harmful, biased, or sensitive content, selection may amplify those properties by prioritizing examples that align with a target reference set or with in-domain structural patterns. In particular, targeted selection can overemphasize narrow behaviors present in the reference samples, which may reduce diversity, increase spurious correlations, or degrade performance on underrepresented groups and topics. As a result, practitioners should treat coreset selection as part of the safety pipeline, auditing candidate pools, filtering toxic or private data, and evaluating for fairness and robustness after selection.

TRIM is not designed to detect or remove unsafe content, nor does it provide guarantees about privacy or bias. The method also does not prevent misuse of instruction tuning to produce models optimized for harmful objectives; it only makes selection more efficient. We recommend using TRIM alongside established dataset governance practices (licensing checks, PII filtering, toxicity screening) and downstream evaluations that include safety-focused benchmarks and red-teaming where appropriate.

Overall, we expect TRIM's primary impact to be enabling more compute-efficient and targeted dataset curation for model adaptation, while emphasizing that responsible deployment requires careful dataset auditing and post-selection evaluation to mitigate risks of bias amplification and harmful content retention.

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

# A. Extended Related Work

This section provides extended details on prior work in data selection for instruction tuning, expanding upon the summary in Section 2.

**Target-Aware Influence-Based Selection**    Influence-based methods aim to quantify the causal effect of a candidate training example on model performance with respect to a target task. A foundational approach, Influence Functions (Koh & Liang, 2017), approximates the impact of upweighting a training point via second-order Hessian computations. As these are infeasible for LLMs, scalable surrogates have been proposed. `DataInf` (Kwon et al., 2024) and `In2Core` (San Joaquin et al., 2024) exploit the low-rank structure of LoRA finetuning to derive closed-form influence approximations, enabling per-sample scoring. `DealREC` (Lin et al., 2024b) instead estimates Hessian-vector products on surrogate models and adjusts with gradient norms as proxies for learning difficulty.

First-order alternatives avoid Hessians by tracking gradient similarity across training. `TracIn` (Pruthi et al., 2020) accumulates dot products of gradients between training and target examples over checkpoints. `LESS` (Xia et al., 2024), designed for instruction tuning, stores compact gradient features with optimizer-aware weighting, yielding strong coresets that can rival full-data finetuning. `BIDS` (Dai et al., 2025) further balances capabilities across tasks by normalizing influence scores prior to selection.

**Training-Dynamics-Based Selection**    Other methods focus on learning dynamics without explicit target sets, seeking data that is broadly informative. `TAGCOS` (Zhang et al., 2025b) clusters gradient features from warmup checkpoints to capture representativeness. `STAFF` (Zhang et al., 2025c) uses gradient norms from smaller surrogate models as effort-based scores.

Classical approaches like `Forgetting` (Toneva et al., 2018), `EL2N`, and `GraNd` (Paul et al., 2021) track prediction changes, error norms, and gradient norms, respectively, during early training. `Moderate` (Xia et al., 2022) score samples by the distance of the sample features to the class median. $\mathbb{D}^2$-`Pruning` (Maharana et al., 2024) uses a graph-based framework that uses the feature density as a measure of the diversity of samples and the prediction variance as the difficulty of the sample. Forward-only methods mitigate the cost of training on all candidates: `S2L` (Yang et al., 2024) leverages small-model losses to guide large-model selection, and `DUAL` (Cho et al., 2025) combines uncertainty with difficulty to stabilize pruning.

**Heuristic and Scorer-Based Selection**    This family includes efficient, task-agnostic methods. `BM25` (Robertson et al., 2009) and `DSIR` (Xie et al., 2023) select based on lexical overlap or $n$-gram distributions, but lack target awareness. Recent works employ lightweight scorers or LLMs-as-judges to filter or rank examples. Examples include filtering low-quality responses (Chen et al., 2024a), ranking samples for diversity (Ge et al., 2024), or targeting difficult instructions (Li et al., 2024; Liu et al., 2024a). These methods are efficient but depend on proxy notions of data quality.

**Optimization-Based Selection**    A principled line of work casts data selection as optimization. `TSDS` (Liu et al., 2024b) minimizes distribution alignment loss between coresets and target distributions. `GREATS` (Wang et al., 2024) greedily selects data by gradient alignment with validation batches. `PDS` (Gu et al., 2025) frames the task as optimal control, linking selection scores to downstream impact. More recently, `QCS` (Chen et al., 2026) formulates coreset construction as a bi-level optimization problem that couples *sequence-level* selection with *token-level* mining. In principle, differentiating through the lower-level token mining step can introduce second-order terms (e.g., Hessian-related hypergradients), and `QCS` proposes a relaxed probabilistic variant to avoid explicit Hessian computation. Overall, while these approaches are conceptually elegant, their optimization loops (and associated higher-order machinery or relaxations) can remain computationally heavy at scale.

**Token-Centric and Attention-Based Selection.**    Recognizing the limitations of sample-level metrics, the most recent work has begun to shift towards token-level signals. These methods leverage the internal mechanisms of Transformers to find important data. For instance, `LADM` uses attention scores to measure long-range dependencies for long-context data selection (Chen et al., 2025). `T-SHIRT` proposes a hierarchical filtering scheme for general-purpose data pruning, rather than for the task-specific, targeted selection that we address (Fu et al., 2026). Its complex pipeline first uses K-Means clustering for a coarse, document-level selection, then refines this by scoring individual tokens via iterative input perturbations, making it computationally intensive.

**Connection to Linguistic Collapse** `Linguistic Collapse` (Wu & Papyan, 2024) extends `Neural Collapse` (Papyan et al., 2020) theory to causal LMs, documenting that with increased scale and appropriate regularization, the final-layer token representations align toward class-like centroids, with equinorm/equiangular structure, and that this geometry correlates with generalization. This effect is characterized at (or near) the end of *pretraining*. Our setting focuses on *instruction finetuning* with relatively shallow training trajectories and task-specific distributions. The relation is therefore tangential but pertinent: (i) *Shared space.* `TRIM` scores examples via token-level hidden-state similarities, precisely where collapse-like geometry manifests; stronger centroidal structure from pretraining can sharpen `TRIM`'s cosine signals for rare, task-informative tokens. (ii) *Local reshaping under finetuning.* Instruction FT can selectively reshape hidden states toward the target format; aggregating scores across warmup checkpoints (LR-weighted) makes `TRIM` less sensitive to any late-stage drift. (iii) *Data selection as geometry control.* By emphasizing `TF-IDF`-weighted, task-specific tokens, `TRIM` may preferentially retain examples that reinforce useful token centroids for the downstream task rather than inducing indiscriminate collapse. A full causal link between collapse metrics during pretraining and FT-time selection effectiveness remains an interesting open question.

**Positioning `TRIM`** `TRIM` contributes to the emerging token-centric paradigm but is differentiated by its specific focus on efficient, *targeted* data selection. Unlike methods for general pruning or long-context analysis, `TRIM` uses a lightweight, single-pass heuristic to find data relevant to a small set of target examples.

Our approach shares a philosophical connection with *token sparsification* methods designed for model efficiency, most notably in computer vision. For example, `DynamicViT` (Rao et al., 2021) introduces a lightweight prediction module to progressively prune uninformative image patch tokens at multiple layers of a Vision Transformer. This drastically reduces FLOPs with minimal accuracy loss by focusing computation on the most salient parts of an input. Both `TRIM` and `DynamicViT` operate on the principle that focusing computation on a small subset of important tokens is a highly effective strategy for efficiency, one applies it to data selection, the other to the inference pass itself.

The specific mechanism of `TRIM` is grounded in prior work on attention analysis. The core idea of using raw attention scores as an explicit, guiding signal, rather than just for representation mixing, has proven successful in other NLP domains. For instance, the work on *Self-Attention Guided Copy Mechanisms* (Xu et al., 2020) used attention scores to inform the model which words from a source text were important enough to be copied directly into a summary, demonstrating their utility as a direct signal of importance. `TRIM` adapts this concept, using attention to signal which tokens are most important for defining a task-specific fingerprint.

Furthermore, our multi-layer approach to calculating salience is inspired by interpretability research. Work on "Quantifying Attention Flow" (Abnar & Zuidema, 2020) established the value of aggregating attention across all layers to build a complete information flow graph, thereby determining a token's overall importance. While their graph-based max-flow algorithm is designed for deep interpretability, `TRIM` adapts the underlying principle into a fast heuristic (focus and centrality) suitable for a large-scale, practical data selection task.

## B. Pseudocode for `TRIM`

We provide detailed pseudocode for the complete `TRIM` pipeline to complement the description in Section 3. The main driver, presented in Algorithm 3, orchestrates the overall workflow: model warmup, fingerprint generation, candidate scoring, and final coreset selection. This algorithm relies on two core subroutines that correspond to the two stages of our method.

**Stage I**, implemented in BUILDFINGERPRINTS (Algorithm 1), details the logic from Section 3.1. It processes the target validation set $\mathcal{T}_{\text{val}}$ using the warmed-up model $\mathbf{M}_{\text{warmup}}$. For each token, it computes the attention-derived salience $\alpha_i$ by combining the director ($Q_i$) and hub ($\tilde{K}_i$) scores. These salience values are then used to create the final, salience-weighted token fingerprints $\mathcal{F} = \{f_t\}$.

**Stage II**, detailed in SCORECANDIDATES (Algorithm 2), executes the scoring process from Section 3.2. For each candidate, it computes token-wise similarity to the fingerprints, applying a penalized backoff for out-of-vocabulary types. These token scores are then aggregated into a robust example score using a mean-max pooling strategy with a coverage bonus.

Together, these algorithms provide a concrete implementation of the forward-only `TRIM` pipeline, culminating in the selection of the coreset $\mathcal{C}$ for downstream finetuning.

---

**Algorithm 1** BUILDFINGERPRINTS (Stage I)

---

**Require:** Warmed model $\mathbf{M}_{\text{warmup}}$, target val $\mathcal{T}_{\text{val}}$, number of layers $L$
**Require:** Salience weights $(w_Q, w_K) = (\frac{1}{2}, \frac{1}{2})$
**Ensure:** Fingerprint dictionary $\mathcal{F} = \{f_t\}$
 1: $\mathcal{F} \leftarrow \emptyset$
 2: **for** each $v \in \mathcal{T}_{\text{val}}$ **do**
 3:    Run $\mathbf{M}_{\text{warmup}}$ once (attn+states, last $L$)
 4:    **for** each position $i$ in $v$ **do**
 5:       Compute $Q_i$ and $\tilde{K}_i$ (eqs. in text)
 6:       $\alpha_i \leftarrow w_Q Q_i + w_K \tilde{K}_i$
 7:    **end for**
 8: **end for**
 9: **for** each token type $t$ seen in $\mathcal{T}_{\text{val}}$ **do**
10:    $O_t \leftarrow \{(v, i) : \text{type}(v_i) = t\}$
11:    $\hat{h}_{v,i} \leftarrow h_{v,i}/\|h_{v,i}\|_2$
12:    $f_t \leftarrow \text{normalize}\Big(\sum_{(v,i)\in O_t} \alpha_{v,i}\hat{h}_{v,i}\Big)$
13:    $\mathcal{F}[t] \leftarrow f_t$
14: **end for**
15: **return** $\mathcal{F}$

---

**Algorithm 2** SCORECANDIDATES (Stage II)

---

**Require:** $\mathbf{M}_{\text{warmup}}$, pool $\mathcal{S}$, fingerprints $\mathcal{F}$
**Require:** Pool weights $(w_\mu, w_m) = (\frac{1}{2}, \frac{1}{2})$, coverage weight $\eta$, nearest-fingerprint fallback penalty $\lambda \in (0,1)$
**Ensure:** Scores $\{S(c)\}_{c\in\mathcal{S}}$
 1: **for** each $c \in \mathcal{S}$ **do**
 2:    Run $\mathbf{M}_{\text{warmup}}$ once to get $\hat{h}_{c,j}$ for all $j$
 3:    $\mathcal{J}(c) \leftarrow \emptyset$
 4:    **for** each position $j$ **do**
 5:       $t_j \leftarrow \text{type}(c_j)$
 6:       **if** $t_j \in \mathcal{F}$ **then**
 7:          $s_j \leftarrow \cos(\hat{h}_{c,j}, \mathcal{F}[t_j])$
 8:       **else**
 9:          $\bar{t}_j \leftarrow \arg\max_{t\in\mathcal{F}} \cos(e_{t_j}, e_t)$
10:          $s_j \leftarrow \lambda \cdot \cos(\hat{h}_{c,j}, \mathcal{F}[\bar{t}_j])$
11:       **end if**
12:       $\mathcal{J}(c) \leftarrow \mathcal{J}(c) \cup \{j\}$
13:    **end for**
14:    **if** $|\mathcal{J}(c)| = 0$ **then**
15:       $S(c) \leftarrow -\infty$
16:    **else**
17:       $\mu \leftarrow \frac{1}{|\mathcal{J}(c)|} \sum_{j\in\mathcal{J}(c)} s_j$
18:       $m \leftarrow \max_{j\in\mathcal{J}(c)} s_j$
19:       $\kappa \leftarrow \frac{|\mathcal{J}(c)|}{|c|}$
20:       $S(c) \leftarrow w_\mu \mu + w_m m + \eta\kappa$
21:    **end if**
22: **end for**
23: **return** $\{S(c)\}$

---

**Algorithm 3** TRIM Framework (Warmup → Stage I → Stage II → Selection)

---

**Require:** Base LLM $\mathbf{M}_0$, pool $\mathcal{S}$, target val $\mathcal{T}_{\text{val}}$, coreset size $K$
**Ensure:** Coreset $\mathcal{C} \subset \mathcal{S}$ of size $K$
 1: **Warmup:** sample $\mathcal{S}_{\text{warmup}} \subset \mathcal{S}$ (∼5%); fine-tune $\mathbf{M}_0$
 2: $\mathbf{M}_{\text{warmup}} \leftarrow$ adapted model
 3: $\mathcal{F} \leftarrow$ BUILDFINGERPRINTS$(\mathbf{M}_{\text{warmup}}, \mathcal{T}_{\text{val}}, L)$
 4: $\{S(c)\} \leftarrow$ SCORECANDIDATES$(\mathbf{M}_{\text{warmup}}, \mathcal{S}, \mathcal{F})$
 5: $\mathcal{C} \leftarrow$ top-$K$ candidates by $S(c)$
 6: **return** $\mathcal{C}$

---

## C. Details in the Algorithm

**Handling Non-fingerprinted Classes**    The Nearest fingerprint fallback (NFF) mapping is designed to handle token classes that do not appear in the target set (and hence lack fingerprints) without introducing per-example nearest-neighbor overhead. Let $\mathcal{V}$ be the vocabulary and $\mathcal{F} \subseteq \mathcal{V}$ the fingerprinted token classes. We unit-normalize all input embeddings $\{e_t\}_{t \in \mathcal{V}}$ from the warmup model and precompute a lookup table

$$\pi : \mathcal{V} \to \mathcal{F} \quad \text{where} \quad \pi(t) = \arg\max_{u \in \mathcal{F}} \cos(e_t, e_u)$$

This precomputation is done once per tokenizer/model (offline) and cached; at scoring time, each non-fingerprinted token $t_j$ incurs only an $O(1)$ lookup to obtain $\bar{t}_j = \pi(t_j)$ and then uses the penalized similarity $s_j = \lambda \cdot \cos(\hat{h}_{c,j}, f_{\bar{t}_j})$.

In practice, this caching makes NFF negligible in the overall selection runtime, and it ensures that unseen token classes contribute conservatively via the penalty $\lambda$ rather than being discarded entirely.

**Selective Token Scoring.**    A key flexibility of TRIM is the ability to selectively fingerprint and score different parts of an example. This scope is a hyperparameter that can be set to *all* tokens, *prompt-only*, or *response-only*. For tasks where the crucial reasoning steps are in the generated answer (e.g., mathematical reasoning in GSM8K), we can restrict scoring to the response. Conversely, for tasks where the complexity lies in the prompt (e.g., COMMONSENSE QA), we can focus only on the prompt. This allows TRIM to target the most informative part of the data for a given domain while scoring fewer tokens for more efficiency.

**Defining the Scored Token Set.**    If selective token scoring is done, then the token at position $j$ is included in $\mathcal{J}(c)$ if and only if it falls within the chosen scoring scope (prompt, response, or all) and is not a special token (e.g., BOS, EOS, PAD). The token's score $s_j$ is calculated as defined above, whether through a direct fingerprint match or NFF backoff.

$$\mathcal{J}(c) = \{\, j : \text{ token } j \text{ is scored} \,\}. \tag{17}$$

If, for a given candidate, this set is empty ($|\mathcal{J}(c)| = 0$), we assign it a score of $S(c) = -\infty$. The robust pooling operation also includes a light coverage term $\kappa(c)$, matching the scoring rule in Eq. (16).

$$\mu(c) \;=\; \frac{1}{|\mathcal{J}(c)|} \sum_{j \in \mathcal{J}(c)} s_j,$$

$$\tau(c) \;=\; \max_{j \in \mathcal{J}(c)} s_j,$$

$$\kappa(c) \;=\; \frac{|\mathcal{J}(c)|}{|c|}.$$

$$S(c) \;=\; w_\mu\, \mu(c) \;+\; w_m\, \tau(c) \;+\; \eta\, \kappa(c). \tag{18}$$

The coverage term lightly rewards candidates where salient matches are not sparse and mitigates length bias. With a small weight $\eta \ll w_\mu, w_m$, the coverage term acts as a tie-breaker, without suppressing genuinely strong single-token spikes.

**IDF-scaled fingerprinting.**    TRIM can optionally incorporate inverse document frequency (IDF) to reduce the influence of ubiquitous tokens when constructing fingerprints. Concretely, for a token class $t$, we scale each occurrence weight by $\text{IDF}(t)$ (computed over the candidate pool), i.e., we replace $\alpha_{v,i}$ with $\alpha_{v,i} \cdot \text{IDF}(t)$ in Eq. (9). This can be helpful when the candidate pool contains substantial shared formatting or "template" text (which otherwise yields many high-saliency but low-specificity matches), while having negligible impact when the task signal is already distributed across many diverse tokens (e.g., MMLU; see Table 13).

## D. Baseline Data Selection Strategies

This section provides a detailed overview of the baseline methods used for comparison in our experiments.

**Random**    This is the simplest baseline, involving uniform random sampling of a subset of the desired size from the full candidate pool without replacement. It serves as a measure of the performance gain achieved by more sophisticated selection strategies.

**BM25 (Robertson et al., 2009)**    `BM25` is a lexical retrieval method that scores candidate examples based on their textual similarity to the target validation set. It uses word frequency statistics, similar to TF-IDF (Sparck Jones, 1972), to rank candidates, prioritizing those with high lexical overlap with the target examples.

**DSIR (Xie et al., 2023)**    Data Selection via Importance Resampling (`DSIR`) is an efficient method that weights candidate examples based on $n$-gram feature overlap with the target validation distribution. A coreset is then formed by resampling from the candidate pool according to these importance weights.

**CLD (Nagaraj et al., 2025)**    Correlation of Loss Differences (`CLD`) is a forward-only method that identifies impactful training data by measuring the alignment between a candidate's training loss trajectory and that of a held-out validation set. The score for each sample is the Pearson correlation between its vector of epoch-to-epoch loss differences and the average vector for the validation set, requiring only per-sample loss values from training checkpoints.

**S2L (Yang et al., 2024)**    Small-to-Large (`S2L`) is a scalable data selection method that leverages a small proxy model to guide selection for a larger target model. It first collects the training loss trajectories of all candidate examples by training the small model. It then clusters these trajectories and performs balanced, uniform sampling from the resulting clusters to form the final coreset.

**RDS (Hanawa et al., 2021)**    Representation-based Data Selection (`RDS`) uses the model's hidden representations as features to score data. In our experiments, we follow the implementation in Xia et al. (2024), which uses the final-layer hidden state of the last token in a sequence as its representation. Candidate examples are then scored based on the cosine similarity of their representation to the average representation of the target validation set.

**LESS (Xia et al., 2024)**    Low-rank gradient Similarity Search (`LESS`) is an optimizer-aware method for targeted data selection. It adapts the classic gradient-similarity influence formulation to work with the Adam optimizer and variable-length instruction data. To remain efficient, `LESS` uses LoRA and random projections to compute a low-dimensional "gradient datastore" from a short warmup training phase. Candidates are scored by the cosine similarity of their gradient features to those of the target validation set, aggregated across several checkpoints.

**TAGCOS (Zhang et al., 2025b)**    Task-Agnostic Gradient Clustered Coreset Selection (`TAGCOS`) is an unsupervised method that uses sample gradients from a warmed-up model as data representations. The method follows a three-stage pipeline: (1) it computes low-dimensional gradient features for each candidate sample; (2) it performs K-means clustering on these gradient features to group similar data; and (3) it applies an efficient greedy algorithm, Orthogonal Matching Pursuit (OMP), to select a representative subset from within each cluster.

## E. Training Datasets Overview

**Training data.**    Our selection pool $S$ is the union of four public instruction-tuning sources, DOLLY, COT, OASST1, and FLAN_v2, totaling 270,679 examples after filtering (Table 6). To unify heterogeneous sources, we format all datasets into a chat-style schema with special tokens (`<|user|>`, `<|assistant|>`) and an end-of-text marker, following the instruction-tuning pipeline of Wang et al. (2023). We obtain the processed datasets from `LESS` repository (Xia et al., 2024). Unless otherwise stated, we do not rebalance per-source proportions prior to selection; methods operate over the pooled corpus $S$ with fixed preprocessing and tokenization settings across all experiments.

*Table 6.* Source training datasets in the selection pool $S$. "#Samples" reflects post-filter counts.

| DATASET | #SAMPLES | BRIEF PURPOSE |
|---|---|---|
| DOLLY | 15,011 | Instruction following; everyday tasks, Q&A, and summaries. |
| COT | 100,000 | Chain-of-thought style prompts and responses. |
| OASST1 | 55,668 | Multi-turn chat-style conversations. |
| FLAN_v2 | 100,000 | Mixture of diverse tasks for instruction finetuning. |
| TOTAL CORPUS | 270,679 | |

# F. Target Datasets & Evaluation Setup

**Evaluation protocol.** We evaluate our method across a diverse set of reasoning and question-answering benchmarks. The main paper reports results on MMLU, BBH, TYDIQA, and GSM8K, while Appendix N additionally includes experiments on COMMONSENSEQA, SOCIALIQA, and HELLASWAG for completeness and comparison with prior work. It is to be noted that $T_{val}$ is disjoint from the Test set.

*Table 7.* Target datasets and evaluation protocol. "#Val" denotes the size of the few-shot target validation set $T_{val}$ used for data selection; "Eval Shots" denotes the number of in-context examples used at test time. Results for MMLU, BBH, TYDIQA, and GSM8K are reported in the main paper, while results for COMMONSENSEQA, SOCIALIQA, and HELLASWAG are included in the appendix.

| DATASET | #TASKS | #VAL | #TEST | EVAL SHOTS | ANSWER TYPE / TASK |
|---|---|---|---|---|---|
| MMLU | 57 | 285 | 18,721 | 5 | Multiple-choice (letter) |
| BBH | 27 | 81 | 920 | 3 | CoT + final answer |
| TYDIQA | 9 | 9 | 1,713 | 1 | Extractive span |
| COMMONSENSEQA | 1 | 5 | 1,221 | 5 | Multiple-choice |
| SOCIALIQA | 1 | 5 | 1,954 | 0 | Multiple-choice |
| HELLASWAG | 1 | 5 | 10,042 | 5 | Multiple-choice (ending) |
| GSM8K | 1 | 10 | 1,319 | 8 | Numeric answer |

All experiments follow a unified OpenAI-style instruction-tuning evaluation protocol (Wang et al., 2023). For data selection, we use a small few-shot target validation set $T_{val}$ for each (sub)task, with the size determined by the few-shot configuration of the benchmark. Specifically, we use five labeled examples per task for most benchmarks and ten for GSM8K, with no access to test labels during selection.

At evaluation time, we use fixed in-context exemplars with the shot counts shown in Table 7, sampled from development pools disjoint from $T_{val}$. We report accuracy for multiple-choice benchmarks, exact match for span-extraction tasks (TYDIQA), and exact match on the final numeric answer for GSM8K. All prompts, decoding parameters, and evaluation settings are held fixed across methods; no task-specific hyperparameters are tuned on test sets.

Datasets are obtained via the `HuggingFace datasets` library (Wolf et al., 2020), and evaluation is performed using the EleutherAI `lm-evaluation-harness` (Gao et al., 2023).

# G. Training Details and Hyperparameters

## G.1. Artifact Licenses

According to their license, all the LLMs used in this paper fall under acceptable use cases. The licenses for the models are linked for perusal: LLAMA-2-7B, LLAMA-2-13B, LLAMA-3.2-1B, LLAMA-3.1-8B, PYTHIA-70M and, MISTRAL 7B (V0.3). All software dependencies, including PyTorch and torchvision, are open-source and distributed under MIT or BSD-compatible licenses.

## G.2. Implementation Details and Shared Hyperparameters

All models in Sections 4.1, 4.2, 4.3, and Section 4.4 are trained with the same setup, closely following LESS (Xia et al., 2024), using an identical training recipe for data loading, preprocessing, optimizer/schedule, precision, and LoRA configuration (outlined in Table 8); only the underlying architecture varies unless otherwise noted.

The implementational details of Section 4.5 is provided in Appendix M.

## G.3. Model Size and Computational Budget

We evaluate a primary backbone model and study transfer across model sizes using a shared training/adaptation recipe (Appendix G.2). Concretely, our main experiments use LLAMA-2-7B ($\approx$7B parameters). For transferability, we apply coresets selected with a LLAMA-2-7B scorer to LLAMA-2-13B ($\approx$13B parameters) and MISTRAL-7B (V0.3) ($\approx$7B parameters). For completeness, Appendix N additionally reports earlier experiments using LLAMA-3.2-1B ($\approx$1B parameters)

*Table 8.* Shared hyperparameters used across all runs; only the architecture varies.

| | |
|---|---|
| Optimizer / LR | AdamW / $2 \times 10^{-5}$ |
| Schedule / Warmup | linear / 0.03 |
| Epochs | 4 |
| Per-device BS / Accum | 2 / 32 (eff. 64) |
| Max sequence length | 2048 |
| Precision | bf16 |
| LoRA (targets) | r=128, $\alpha = 512$, $p = 0.1$ (`q,k,v,o_proj`) |
| Eval / Saves | no eval; save/1055, keep 15 |

and LLAMA-3.1-8B ($\approx$8B parameters).

All runs were executed on a single NVIDIA H200 GPU on an internal cluster (CUDA 12.1, PyTorch 2.1, HuggingFace Transformers 4.46). We report wall-clock time or GPU-hours in Section 5.

## H. Hyperparameter Ablations on MMLU

We ablate key design choices in TRIM on MMLU while holding the rest of the selection and fine-tuning pipeline fixed to the main setup. Unless stated otherwise, each configuration selects a 5% coreset from the same candidate pool and fine-tunes the same base model as in the corresponding MMLU experiment. We report MMLU test accuracy (%; mean $\pm$ std. over 3 seeds).

**Default configuration.** Unless otherwise noted, we use the following TRIM configuration for MMLU: saliency aggregated over the last $L$=6 layers; robust pooling with $0.5 \cdot$ MEAN$+0.5 \cdot$ MAX$+0.05 \cdot$ COVERAGE (Appendix C); no IDF reweighting; an NFF penalty of $\lambda$=0.9; and scoring using only the last checkpoint. Default values are highlighted in each table.

### H.1. Number of layers used for saliency aggregation

In Table 9, we vary the number of final transformer layers $L$ whose attention-derived saliency signals are aggregated when building token fingerprints.

*Table 9.* MMLU ablation: number of last layers $L$ used for saliency aggregation. Default value highlighted.

| # Layers ($L$) | MMLU Accuracy (%) |
|---|---|
| Last 3 | 48.78 *(0.3)* |
| Last 6 | 49.33 *(0.4)* |
| Last 12 | 49.43 *(0.2)* |

### H.2. Layer group used for fingerprinting

In Table 10, we study the effect of constructing fingerprints from different groups of layers in the warmup model. Our default choice of using the last layers is motivated by prior evidence that layer contributions in LLMs are not uniform. In particular, analyses of Llama-2 representations suggest that lower layers tend to encode more lexical and local semantic information, whereas higher layers are more closely tied to next-token prediction and final decision-making (Liu et al., 2024c; Gao et al., 2024). Therefore, rather than assuming that all layers contribute equally, we explicitly ablate fingerprints constructed from early, middle, and late layers. The results show that the last six layers provide the strongest target-conditioned fingerprints across both MMLU and GSM8K, supporting our use of semantically mature upper-layer representations for candidate selection.

### H.3. Saliency weighting for fingerprinting

In Table 11, we study the effect of varying the relative weights assigned to the row and column saliency scores used during fingerprint construction (see Equation (7)). The row and column terms capture complementary views of token importance:

*Table 10.* MMLU and GSM8K ablation: layer group used for fingerprint construction. Default value highlighted.

| Layer Group | MMLU Accuracy (%) | GSM8K Accuracy (%) |
|---|---|---|
| Early 6 layers | 48.56 *(0.2)* | 20.34 *(0.6)* |
| Middle 6 layers | 48.14 *(0.4)* | 24.23 *(0.4)* |
| Last 6 layers | 49.33 *(0.4)* | 29.52 *(0.3)* |

the row saliency reflects how a token distributes attention to the rest of the sequence, while the column saliency reflects how much the token is attended to by other tokens. Accordingly, our default setting assigns equal weight to both terms. The ablation shows that balanced or near-balanced weighting performs best, while relying exclusively on either row or column saliency leads to lower accuracy.

*Table 11.* MMLU ablation: row and column saliency weighting used for fingerprinting. Default value highlighted.

| Saliency Weights $(w_Q, w_K)$ | MMLU Accuracy (%) |
|---|---|
| (0, 1) | 47.24 |
| (0.1, 0.9) | 47.83 |
| (0.25, 0.75) | 49.32 |
| (0.5, 0.5) | 49.33 |
| (0.75, 0.25) | 49.21 |
| (0.9, 0.1) | 48.34 |
| (1, 0) | 47.43 |

## H.4. Pooling function for candidate scoring

In Table 12, we compare pooling choices used to aggregate token-level matches into a sample-level score, including mean pooling, max pooling, and coverage-based pooling (Appendix C).

*Table 12.* MMLU ablation: pooling function used for candidate scoring. Default value highlighted.

| Pooling | MMLU Accuracy (%) |
|---|---|
| Mean only | 48.26 *(0.1)* |
| Max only | 49.14 *(0.3)* |
| Mean $+ 0.05 \cdot$Coverage | 48.75 *(0.3)* |
| $0.5 \cdot$Mean $+ 0.5 \cdot$Max | 48.75 *(0.1)* |
| Robust: $0.5 \cdot$Mean $+ 0.5 \cdot$Max $+ 0.10 \cdot$Coverage | 48.77 *(0.2)* |
| Robust: $0.5 \cdot$Mean $+ 0.5 \cdot$Max $+ 0.05 \cdot$Coverage | 49.43 *(0.2)* |

## H.5. IDF reweighting

In Table 13, we evaluate whether inverse document frequency (IDF) reweighting improves robustness by down-weighting frequent tokens and emphasizing rarer, potentially more informative tokens.

*Table 13.* MMLU ablation: effect of IDF reweighting. Default value highlighted.

| Setting | MMLU Accuracy (%) |
|---|---|
| No IDF | 49.43 *(0.2)* |
| IDF (on) | 49.31 *(0.1)* |

### H.6. Penalizing non-fingerprinted similarities

In Table 14, we ablate the fallback penalty $\lambda$ in Equation (12), which controls how strongly we penalize matches involving token classes that are not observed in the target validation set and must be mapped to a nearest fingerprinted class.

*Table 14.* MMLU ablation: effect of the NFF penalty $\lambda$. Default value highlighted.

| $\lambda$ | MMLU Accuracy (%) |
|---|---|
| No NFF | 44.79 (0.45) |
| 0.5 | 46.04 (0.8) |
| 0.9 | 49.43 (0.2) |
| 1.00 | 48.58 (0.5) |

### H.7. Checkpoints used for scoring

In Table 15, we ablate the number of checkpoints used during scoring. When using multiple checkpoints, we aggregate per-checkpoint scores using learning-rate scaling, following the convention in (Xia et al., 2024).

*Table 15.* MMLU ablation: checkpoints used in scoring. Default value highlighted.

| Checkpoint setting | MMLU Accuracy (%) |
|---|---|
| Pretrained model | 44.26 (0.1) |
| Earliest warmup checkpoint | 46.37 (0.2) |
| Last warmup checkpoint | 49.43 (0.2) |
| All warmup checkpoints | 49.43 (0.5) |

## I. Bias and Domain-Shift Robustness of Attention-Derived Saliency

To probe whether attention-derived saliency overemphasizes stylistic or high-frequency tokens in a way that harms generalization, we evaluate models fine-tuned on 5% TRIM-selected coresets (targeted to GSM8K) on additional benchmarks beyond GSM8K. Table 16 reports results on MMLU and TRUTHFULQA (MC2).

*Table 16.* Out-of-task evaluation on MMLU and TRUTHFULQA (MC2) for LLAMA-2-7B. Coresets are 5% selected from the DOLLY/FLAN_V2/COT/OASST1 pool using GSM8K targets (10 examples). Mean accuracy (%) with std. over 3 runs.

| Fine-tuning setting | MMLU | TRUTHFULQA (MC2) |
|---|---|---|
| Pretrained | 45.6 | 39.2 |
| LESS 5% coreset | 46.30 (0.4) | 39.91 (0.2) |
| TRIM 5% coreset | 46.12 (0.3) | 39.56 (0.2) |

Importantly, for MMLU and TRUTHFULQA all methods still fine-tune on 5% drawn from the same underlying instruction pool; GSM8K supervision only determines *which* 5% is selected, not the candidate distribution itself. Accordingly, a GSM8K-targeted 5% coreset need not be optimal for MMLU/TRUTHFULQA, but it should not systematically degrade performance if the selection procedure introduces no harmful biases. Consistent with this expectation, TRIM-selected coresets do not reduce performance relative to the pretrained model and remain comparable to LESS on both benchmarks, while modestly improving over the pretrained baseline.

## J. Robustness to Target Validation Sets

**Setup.** All experiments in this section use the same selection setup: we select 5% coresets from the DOLLY/FLAN_V2/COT/OASST1 candidate pool using GSM8K target validation examples, fine-tune LLAMA-2-7B using LoRA, and evaluate on GSM8K under a fixed 8-shot protocol. Results are mean accuracy (%) with std. over 3 seeds.

## J.1. Robustness to validation set size

To quantify TRIM's sensitivity to the size of the target validation set, we vary the number of target examples used to construct token fingerprints. Table 17 reports GSM8K accuracy as the target set size increases.

*Table 17.* Effect of target validation set size on GSM8K (8-shot). 5% coreset selected from DOLLY/FLAN_V2/COT/OASST1. Mean accuracy (%) with std. over 3 runs. Default value highlighted.

| # Target examples | GSM8K accuracy (%) |
|---|---|
| 5 | 27.37 (0.1) |
| 8 | 28.43 (0.2) |
| 10 | 29.52 (0.3) |
| 20 | 30.25 (0.2) |
| 50 | 29.84 (0.1) |
| 100 | 30.18 (0.4) |

Performance improves as the validation set grows: increasing the number of target examples exposes TRIM to a broader set of token contexts, yielding more informative fingerprints. Beyond roughly 20–50 examples, gains saturate and fluctuations remain within the reported standard deviations, indicating that TRIM does not require large target sets to be effective. Even 100 target examples remain negligible relative to the candidate pool, so the added compute and memory overhead is minor in practice.

## J.2. Robustness to noisy targets

We further evaluate robustness by injecting noise into the target examples. Specifically, for a randomly chosen fraction of target examples, we swap their answer prompts with those of other validation examples. Table 18 reports GSM8K performance as the noise ratio increases.

*Table 18.* Robustness to noisy targets on GSM8K (8-shot). Targets: 10 examples. We vary the fraction of noisy target samples used to construct fingerprints. Mean accuracy (%) with std. over 3 runs. Default value highlighted.

| Noisy percentage (%) | GSM8K accuracy (%) |
|---|---|
| 0 | 29.52 (0.3) |
| 10 | 29.65 (0.1) |
| 25 | 29.45 (0.4) |
| 50 | 28.51 (0.1) |
| 75 | 27.80 (0.1) |
| 100 | 27.29 (0.1) |

Accuracy declines gradually with increasing noise, but the degradation is not catastrophic: even at 100% noise, performance remains comparable to the small-target regime in Table 17. Intuitively, noisy targets still induce attention over similar token support (tokens remain drawn from the same underlying pool), so aggregate token statistics are only partially perturbed. The primary effect of noise is to distort the *relative weighting* between the query- and key-based saliency components (our $Q/K$ views): mismatched question–answer pairs lead to less task-aligned attention patterns, and the combined $w_Q - w_K$ aggregation becomes less informative. Nevertheless, because TRIM aggregates saliency across many tokens and layers, the resulting fingerprints remain reasonably stable, leading to only moderate performance loss even under substantial noise.

## K. Source-Subset Composition of Selected Coresets

Figure 4 summarizes how different selection methods draw from the mixed instruction pool (COT, FLAN_V2, OASST1, DOLLY) when constructing 5% coresets for MMLU, BBH, TYDIQA, and GSM8K. The methods exhibit markedly different preferences over sources, but these preferences do not translate into a simple "more from subset ⇒ better performance"

relationship. This indicates that all sources contain valuable examples and that the main challenge is identifying the most task-relevant instances.

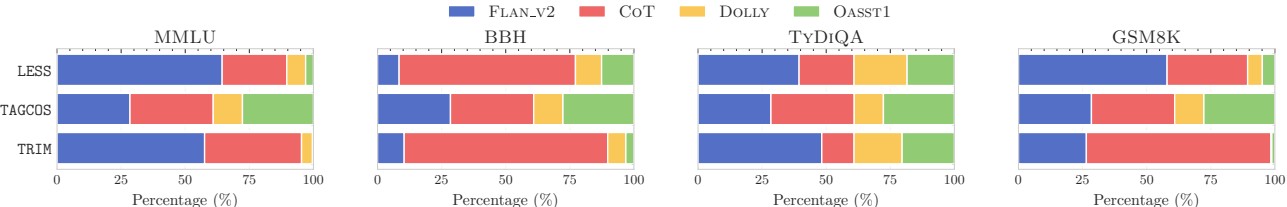

*Figure 4.* Distribution of selected examples across training subsets for different methods and target tasks. Each bar shows the percentage breakdown of data selected from COT, DOLLY, FLAN_V2, and OASST1 datasets.

A clear distinction emerges between task-agnostic and task-adaptive selection. TAGCOS produces an almost unchanged mixture across targets, consistent with its task-agnostic design. In contrast, TRIM and LESS adapt their selection distributions to the target task, but they do so using different signals: TRIM relies on *token-level* representational matching, whereas LESS operates at the *sequence level* using per-example training dynamics.

Qualitatively, TRIM tends to emphasize reasoning-centric sources when the target requires explicit multi-step inference (e.g., BBH and GSM8K), while leaning more on broad instruction-following/knowledge-heavy sources for MMLU. For TYDIQA, TRIM adopts a more mixed allocation, consistent with multilingual QA benefiting from diversity in instruction style and linguistic coverage rather than a single dominant format. LESS also exhibits target-dependent shifts, but as a sequence-level method it is sensitive to whole-example training signals (rather than localized token overlaps), which can induce different subset preferences even when both methods are task-adaptive.

## L. Qualitative Inspection of Token Fingerprints and Nearest-Fingerprinted Fallback (NFF)

**What is shown.** We present qualitative examples that make TRIM concrete: (i) representative *fingerprinted tokens* (token classes that appear in the target validation set and thus receive fingerprints) in Table 19, and (ii) examples of *Nearest-Fingerprinted Fallback (NFF)*, where a token without a fingerprint is mapped to the nearest fingerprinted token in input-embedding space and scored using the mapped fingerprint (with an NFF penalty factor, when enabled) in Table 20.

*Table 19.* Example fingerprinted tokens (decoded strings) for each target task.

| Target task | Fingerprinted token (decoded) |
|---|---|
| MMLU | `"solution"` |
| | `"intelligence"` |
| | `"African"` |
| BBH | `"Western"` |
| | `"Furthermore"` |
| | `"highest"` |
| TYDIQA | `"ju"` |
| | `"‘ds‘"` |
| | `"‘ya‘"` |
| GSM8K | `"+"` |
| | `"half"` |
| | `"Since"` |

**Takeaway.** Across target tasks, the fingerprinted token set contains task-relevant lexical and formatting patterns under the chosen token scope. When a token does not receive a fingerprint, *Nearest-Fingerprinted Fallback (NFF)* provides a simple backoff by mapping to a nearby fingerprinted token in embedding space.

*Table 20.* Nearest-Fingerprinted Fallback (NFF) examples. For a token without a fingerprint, NFF maps it to the nearest fingerprinted token in input-embedding space and reuses the mapped fingerprint for scoring (optionally down-weighted by the NFF penalty).

| Token w/o fingerprint | NFF mapped fingerprint token |
|---|---|
| `"solutions"` | `"solution"` |
| `"location"` | `"sites"` |
| `"players"` | `"games"` |
| `"may"` | `"might"` |
| `"costs"` | `"bill"` |
| `"paid"` | `"bought"` |

## M. In-Domain Selection with No Target Samples

**Setting overview.** We study in-domain selection when there is no external target dataset: the candidate pool itself defines the domain, and the goal is to select a coreset that preserves the pool distribution. We follow the specialized math selection setting of S2L (Yang et al., 2024) using MATHINSTRUCT (Yue et al., 2024), a 262K-example instruction-tuning corpus aggregated from 13 math sources. To enable validation-driven selectors without introducing a separate target dataset, we construct a small in-domain *target pool* $\mathcal{T}_{\text{val}}$ via stratified sampling across sources and use it consistently for all methods that require target/validation examples (e.g., TRIM, LESS). We evaluate this setup with the MAMMOTH/MATHINSTRUCT evaluation pipeline and report exact-match accuracy, consistent with S2L.

**Data format and training objective.** MATHINSTRUCT unifies heterogeneous math datasets into an follow an Alpaca-style instruction-following format (Chen et al., 2024b), where each example is an `instruction` (with optional `input`) paired with a `response`. For supervised fine-tuning, we use standard next-token prediction with the loss masked on the instruction portion, training the model to predict only the response tokens.

**Reference target pool for fingerprinting ($\mathcal{T}_{\text{val}}$).** Let the 13 MATHINSTRUCT sources be $\{s_k\}_{k=1}^{13}$. We build $\mathcal{T}_{\text{val}}$ by sampling $m$ examples uniformly at random from each source, yielding $|\mathcal{T}_{\text{val}}| = 13m$. Unless otherwise stated, we use $m=20$ (i.e., $|\mathcal{T}_{\text{val}}|=260$). As stated before, $\mathcal{T}_{\text{val}} \subset \mathcal{S}$ by construction. Results are averaged over multiple runs, indicating that a randomly selected $\mathcal{T}_{\text{val}}$ is sufficient for identifying high impact data samples.

**Evaluation protocol and benchmarks.** We evaluate on the same six open-form math benchmarks used by S2L under the 0-shot MAMMOTH evaluation pipeline: GSM8K (Cobbe et al., 2021), MATH (Hendrycks et al., 2021), NUMGLUE (Mishra et al., 2022), SVAMP (Patel et al., 2021), MATHEMATICS (Davies et al., 2021), and SIMULEQ (Koncel-Kedziorski et al., 2016).

Evaluation uses a PoT → CoT fallback prompting scheme (attempt program generation/execution first; otherwise fall back to "think step-by-step") and reports exact match accuracy. Some benchmarks have *source overlap* with MATHINSTRUCT (e.g., GSM8K/MATH/NUMGLUE), while others have *no source overlap* (e.g., SVAMP/MATHEMATICS/SIMULEQ).

**Scoring and training pipeline.** Across all methods, the *evaluation model* is a pretrained LLAMA-2-7B that is fine-tuned on the selected subset using the same fixed-epochs protocol. Selection scores are computed using one of two scorer recipes: (i) a warm-started LLAMA-2-7B scorer (our main-paper default), or (ii) an S2L-style fully-trained proxy scorer (PYTHIA-70M). We summarize the S2L-style proxy training hyperparameters in Table 21.

*Table 21.* S2L-style proxy training hyperparameters on MATHINSTRUCT (used when training the Pythia proxy to collect loss trajectories).

| | |
|---|---|
| Proxy model | PYTHIA-70M |
| Learning rate | $2 \times 10^{-5}$ |
| Batch size (global) | 128 |
| Max sequence length | 512 |
| LR schedule / warmup | cosine / 3% |

**Performance comparison.** Table 22 compares the performance of a LLAMA-2-7B model fine-tuned on 50% coresets selected by various methods using the warm-started LLAMA-2-7B scorer.

*Table 22.* Extended evaluation on six open-form math benchmarks for LLAMA-2-7B fine-tuned on 50% coresets (mean over 3 seeds; std. in parentheses).

| Coreset Method | Benchmarks with Source Overlap | | | | Benchmarks with No Source Overlap | | | |
|---|---|---|---|---|---|---|---|---|
| | GSM8K | MATH | NUMGLUE | Avg | SVAMP | MATHEMATICS | SIMULEQ | Avg |
| *Pretrained (no Fine-tuning)* | 3.10 | 4.20 | 16.50 | 7.93 | 12.40 | 8.30 | 2.30 | 7.67 |
| *Full-data Fine-tuning* | 51.15 *(0.2)* | 29.45 *(0.5)* | 59.34 *(0.2)* | 46.65 | 66.12 *(0.1)* | 42.10 *(0.1)* | 49.50 *(0.3)* | 52.57 |
| Random | 48.16 *(0.5)* | 18.22 *(0.3)* | 55.12 *(0.4)* | 40.50 | 63.30 *(1.2)* | 38.75 *(0.2)* | 45.16 *(0.3)* | 49.07 |
| LESS | 52.10 *(0.1)* | 32.20 *(0.1)* | 60.30 *(0.2)* | 48.20 | 65.45 *(0.2)* | 41.80 *(0.1)* | 48.75 *(0.1)* | 52.00 |
| S2L | 52.45 *(0.1)* | 32.10 *(0.2)* | 60.10 *(0.1)* | 48.22 | 65.30 *(0.2)* | 41.92 *(0.3)* | 49.26 *(0.1)* | 52.16 |
| TAGCOS | 50.20 *(0.2)* | 28.28 *(0.1)* | 58.31 *(0.6)* | 45.60 | 64.15 *(0.4)* | 40.84 *(0.1)* | 48.85 *(0.2)* | 51.28 |
| TRIM **(Ours)** | **52.23** *(0.2)* | **31.40** *(0.3)* | **60.52** *(0.1)* | **48.05** | **65.12** *(0.4)* | **42.13** *(0.4)* | **48.75** *(0.2)* | **52.00** |

**Ablation on target-pool size.** Table 23 varies the number of target samples $m$ per source used to construct $\mathcal{T}_{val}$, holding all other components fixed.

*Table 23.* Ablation on the size of the target pool $\mathcal{T}_{val}$ used for fingerprinting/validation. We sample $m \in \{1, 4, 10, 20\}$ examples per source (13 sources), so $|\mathcal{T}_{val}| = 13m$. We report exact-match accuracy (%) on six open-form math benchmarks (mean over 3 seeds; std. in parentheses).

| $m$ **per source** | $|\mathcal{T}_{val}|$ | Benchmarks with Source Overlap | | | | Benchmarks with No Source Overlap | | | |
|---|---|---|---|---|---|---|---|---|---|
| | | GSM8K | MATH | NUMGLUE | Avg | SVAMP | MATHEMATICS | SIMULEQ | Avg |
| 1 | 13 | 46.20 *(0.3)* | 17.40 *(0.1)* | 52.12 *(0.3)* | 38.57 | 62.14 *(0.4)* | 39.23 *(0.1)* | 44.00 *(0.4)* | 48.46 |
| 4 | 52 | 49.51 *(0.1)* | 24.20 *(0.3)* | 55.47 *(0.4)* | 43.06 | 62.43 *(0.2)* | 40.41 *(0.2)* | 45.16 *(0.1)* | 49.33 |
| 10 | 130 | 51.89 *(0.4)* | 30.10 *(0.6)* | 59.18 *(0.3)* | 47.06 | 65.42 *(0.2)* | 40.57 *(0.1)* | 47.25 *(0.1)* | 51.08 |
| 20 | 260 | 52.23 *(0.2)* | 31.40 *(0.3)* | 60.52 *(0.1)* | 48.05 | 65.12 *(0.4)* | 42.13 *(0.4)* | 48.75 *(0.2)* | 52.00 |

**Ablation on scorer recipe.** Finally, Table 24 compares selection scores computed using (i) the warm-started LLAMA-2-7B scorer and (ii) the fully-trained PYTHIA-70M proxy, while fine-tuning the same pretrained LLAMA-2-7B evaluation model under the fixed-epochs protocol.

*Table 24.* Ablation on the scorer/reference-model recipe used to compute selection scores. Both settings use the same $\mathcal{T}_{val}$ construction (default $m$=20 per source) and fine-tune the same pretrained LLAMA-2-7B evaluation model under the fixed-epochs protocol (mean over 3 seeds; std. in parentheses).

| Selector recipe for scoring | Benchmarks with Source Overlap | | | | Benchmarks with No Source Overlap | | | |
|---|---|---|---|---|---|---|---|---|
| | GSM8K | MATH | NUMGLUE | Avg | SVAMP | MATHEMATICS | SIMULEQ | Avg |
| Warm-start LLAMA-2-7B | 52.23 *(0.2)* | 31.40 *(0.3)* | 60.52 *(0.1)* | 48.05 | 65.12 *(0.4)* | 42.13 *(0.4)* | 48.75 *(0.2)* | 52.00 |
| Fully-trained PYTHIA-70M | 53.40 *(0.1)* | 30.89 *(0.1)* | 60.46 *(0.3)* | 48.25 | 64.89 *(0.2)* | 41.78 *(0.2)* | 47.50 *(0.2)* | 51.39 |

# N. Supplementary Experiments on LLAMA-3.2-1B

**Scope.** In addition to the main-paper experiments (which use LLAMA-2-7B as the primary backbone and study transfer to LLAMA-2-13B and MISTRAL-7B (V0.3)), we report here supplementary results from an earlier evaluation suite using LLAMA-3.2-1B. These results follow the same warmup → score → select → finetune pipeline and are included to demonstrate that the trends observed in the main paper are consistent across model families and benchmark choices.

### N.1. Setup: Commonsense Multiple-Choice Benchmarks

We evaluate on three standard commonsense multiple-choice benchmarks: COMMONSENSEQA (Talmor et al., 2019), SOCIALIQA (Sap et al., 2019), and HELLASWAG (Zellers et al., 2019). We use the same OpenAI-style instruction-tuning

evaluation protocol (Wang et al., 2023) and evaluate using the EleutherAI `lm-evaluation-harness` (Gao et al., 2023). We report accuracy on all tasks. Selection uses a small target validation set $T_{\text{val}}$, and evaluation uses fixed in-context exemplars with shot counts consistent with Table 7; evaluation exemplars are drawn from development pools disjoint from $T_{\text{val}}$. All prompts and decoding settings are held fixed across methods.

### N.2. Budgeted Accuracy on LLAMA-3.2-1B

**Baselines.** We compare against the same families of coreset methods as in the main paper (Appendix D), and all methods use the standardized warmup-and-selection pipeline (Appendix G).

*Table 25.* Accuracy (%) for coreset methods on LLAMA-3.2-1B with a 5% coreset. Results are means over 3 seeds (standard deviations in parentheses). The final column is the macro-average across tasks.

| Coreset Method | COMMONSENSEQA | SOCIALIQA | HELLASWAG | AVERAGE MEAN |
|---|---|---|---|---|
| *Pretrained (no Fine-tuning)* | 29.32 | 42.86 | 48.20 | 40.12 |
| *Full-data Fine-tuning* | 48.24 (1.2) | 45.04 (0.7) | 48.55 (0.7) | 47.27 |
| Random | 34.05 (1.2) | 43.84 (0.1) | 48.21 (0.1) | 42.03 |
| BM25 (Robertson et al., 2009) | 38.88 (0.7) | 44.41 (0.3) | 48.76 (0.1) | 44.02 |
| DSIR (Xie et al., 2023) | 37.16 (0.7) | 44.38 (0.5) | 48.75 (0.2) | 43.43 |
| S2L (Yang et al., 2024) | 34.10 (0.1) | 43.32 (0.2) | 48.57 (0.2) | 41.99 |
| RDS (Hanawa et al., 2021) | 36.16 (1.1) | 43.77 (0.1) | 48.43 (0.3) | 42.79 |
| CLD (Nagaraj et al., 2025) | 33.10 (0.4) | 43.25 (0.5) | 48.35 (0.1) | 41.56 |
| LESS (Xia et al., 2024) | 39.10 (0.7) | 44.52 (0.3) | 49.01 (0.1) | 44.21 |
| TAGCOS (Zhang et al., 2025b) | 34.72 (0.8) | 43.70 (0.2) | 48.76 (0.1) | 42.39 |
| TRIM **(Ours)** | **40.76** (0.6) | **46.26** (0.3) | **49.08** (0.1) | **45.37** |

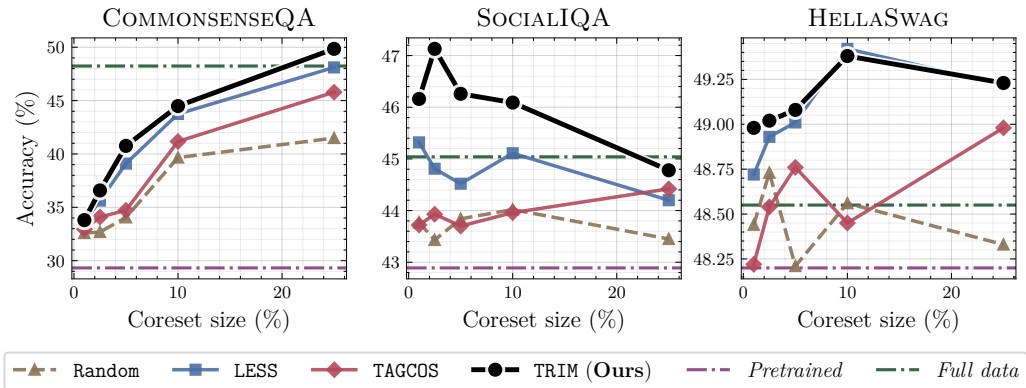

*Figure 5.* TRIM vs. top baselines on LLAMA-3.2-1B across coreset budgets.

**Results and Takeaways.** As summarized in Table 25, TRIM achieves the strongest overall performance under a fixed 5% budget, matching the main-paper trend that token-centric selection yields more accurate coresets than sample-level baselines. Moreover, Figure 5 shows that TRIM's advantage is stable across coreset sizes, indicating robustness to budget choice and supporting its use as a reliable selector under varying compute constraints.

### N.3. Transferability from a Small Scorer

We assess whether a small scorer can curate data that transfers to larger models. Using LLAMA-3.2-1B as the scorer, we select a single 5% coreset and reuse it to fine-tune two larger target models: LLAMA-3.1-8B and MISTRAL-7B (v0.3). We compare (i) TRIM-Transfer, the coreset chosen by the 1B scorer, and (ii) TRIM-Oracle, a coreset selected using the

target model itself as the scorer.

*Table 26.* **Cross-Model and Cross-Scale Transferability of Coresets Selected by a 1B Scorer.** All scores are the mean Accuracy (%) across three seeds. Datasets are: COMMONSENSEQA (CSQA), SOCIALIQA (SIQA), and HELLASWAG (HS).

| Coreset Method | LLAMA-3.1-8B | | | | MISTRAL-7B (v0.3) | | | |
| --- | --- | --- | --- | --- | --- | --- | --- | --- |
| | CSQA | SIQA | HS | AVG. | CSQA | SIQA | HS | AVG. |
| *Pretrained (no Fine-tuning)* | 74.61 | 47.03 | 60.96 | 60.87 | 70.84 | 45.80 | 50.76 | 55.80 |
| `Random` | 75.21 *(0.3)* | 49.13 *(0.1)* | 63.45 *(0.2)* | 62.60 | 74.00 *(0.6)* | 50.59 *(0.2)* | 64.02 *(0.3)* | 63.20 |
| `TRIM-Transfer` (from 1B) | 76.38 *(0.9)* | 51.95 *(0.5)* | 64.19 *(0.1)* | 64.17 | 75.98 *(0.9)* | 54.84 *(0.8)* | 65.35 *(0.2)* | 65.39 |
| `TRIM-Oracle` (in-model) | 75.40 *(0.2)* | 51.11 *(0.9)* | 63.93 *(0.1)* | 63.48 | 75.03 *(0.4)* | 54.12 *(0.4)* | 65.34 *(0.1)* | 64.83 |

**Results and Takeaways.** Table 26 shows that coresets selected by the LLAMA-3.2-1B scorer transfer with high fidelity to both a larger in-family model and an out-of-family target. Consistent with the main-paper findings, `TRIM-Transfer` closely tracks (and in some cases matches) the `TRIM-Oracle` reference, suggesting that `TRIM`'s token-level fingerprints capture signals that generalizes across model scale and architecture.

### N.4. Subset Distribution of Selected Examples

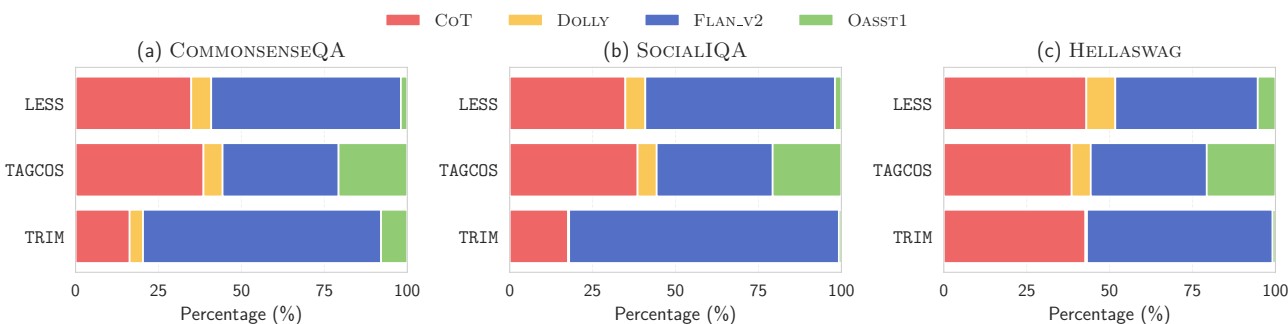

*Figure 6.* Distribution of selected examples across training subsets for different methods and target tasks in the LLAMA-3 supplementary setup. Each bar shows the percentage breakdown of selected data from COT, DOLLY, FLAN_V2, and OASST1.

We present the distribution of selected examples across different data selection methods under the LLAMA-3 evaluation setup in Figure 6.

`TAGCOS`, being a task-agnostic method, maintains nearly identical selection distributions across all four tasks in this setting (approximately 38.5% COT, 35.0% FLAN_V2, 20.7% OASST1, and 5.8% DOLLY). In contrast, both `TRIM` and `LESS` demonstrate task-adaptive behavior, substantially varying their selection patterns based on the specific characteristics of each target task. In our observations for the LLAMA-3 setup, we find that `TRIM` predominantly selects data from the FLAN_V2 dataset for both SOCIALIQA (81.3%) and COMMONSENSEQA (71.8%). This preference is intuitive, as both tasks involve question answering with multiple-choice or short-answer formats, which are well represented in FLAN_V2's diverse instruction-following data. For HELLASWAG, `TRIM` adopts a more balanced approach, selecting comparable amounts from FLAN_V2 (56.0%) and COT (42.7%), consistent with HELLASWAG's commonsense inference format, which benefits from both broad instruction-following and explicit reasoning demonstrations.

