# OpenReview forum: "TRIM: Token-wise Attention-Derived Saliency for Data-Efficient Instruction Tuning"
_ICML.cc/2026/Conference — ICML 2026 regular_

### Official Review · Reviewer_iyQr · 2026-03-10

**Soundness:** 3
**Presentation:** 2
**Significance:** 3
**Originality:** 2
**Overall Recommendation:** 4
**Confidence:** 3

**Summary:**

This paper proposes TRIM, a forward-only token-centric framework for LLM instruction tuning coreset selection, addressing the high cost and inherent biases of sample-level methods. TRIM constructs attention-derived token fingerprints from a small target set to score and rank candidates, and extensive experiments show it outperforms SOTA by up to 9% with a 5% coreset, mitigates length bias, is highly efficient, and validates cross-model transfer and in-domain selection effectiveness with solid ablations.

**Compliance With Llm Reviewing Policy:**

Affirmed.

**Final Justification:**

After the review and rebuttal process, I lean to accept this work, and keep the rating as 4.

**Key Questions For Authors:**

1. Add appendix ablation on layer-wise saliency contributions (early/middle vs. last layers).
2. Supplement with a small experiment on TRIM’s performance with label/formatting target noise.
3. Add a brief evaluation on one long-context instruction tuning benchmark (e.g., LongDocQA).
4. Include 1-2 qualitative case studies of token fingerprints for key tasks (GSM8K/BBH).

**Limitations:**

Please refer to the weakness and questions raised above.

**Strengths And Weaknesses:**

Strengths
1. Novel token-centric design: Shifts from sample-level to fine-grained token-level representation matching, fundamentally addressing length and loss-centric biases of prior work, a key innovation for instruction tuning data curation.
2. Superior efficiency: Forward-only architecture avoids costly gradient/backward computations, achieving orders-of-magnitude higher efficiency than gradient-based SOTA, with rigorous complexity and runtime validation.
3. Strong empirical results: Consistently outperforms all baselines across benchmarks; stands out on low-overlap GSM8K (nearly matching full-data tuning with 5% coreset) and shows robust cross-model/scale transfer.

Weakness
1. Limited layer-specific analysis: Aggregates saliency from last L layers but lacks analysis of why these layers are optimal or lower/middle layer contributions.
2. No long-context task evaluation: Experiments focus on short/medium-context benchmarks, missing validation for long-context instruction tuning (a key LLM use case).
3. Weak fingerprint interpretability: Only minimal qualitative analysis of token fingerprints; lacks visualizations/case studies of how they capture task-specific features.

---

> ### Author Rebuttal · Authors · 2026-03-30
>
> We thank the reviewer for providing insightful feedback and helping us strengthen the paper. We address the questions asked in the review below.
> 1. Our choice to aggregate saliency from the last $L$ layers was motivated by prior evidence that layer contributions in LLMs are not uniform. In particular, prior analysis of Llama-2 representations shows that lower layers tend to encode more lexical/local semantics, whereas higher layers are more directly tied to next-token prediction and final decision-making [1, 2]. At the same time, pruning and layer-quality studies indicate that layer importance is not uniform across depth and can vary across tasks, which further motivates explicit empirical validation rather than assuming all layers contribute equally [3,4]. To address this directly, we conducted a layer-group ablation comparing TRIM fingerprints built from early layers $(0-5)$, middle layers $(13-18)$, and late layers $(26-31)$ and show the results in Table R4. This ablation will help clarify that TRIM does not rely on an assumption of uniform layer utility; rather, it leverages the empirically motivated intuition that semantically mature upper-layer representations are more informative for target-conditioned data selection.
>
> **Table R4: Performance of  Llama 2 7 B finetuned on 5% coresets**
>
> | Layer group     |        MMLU | GSM8k       |
> | --------------- | ----------: | ----------- |
> | Early 6 layers  | 48.56 (0.2) | 20.34 (0.6) |
> | Middle 6 layers | 48.14 (0.4) | 24.23 (0.4) |
> | Last 6 layers   | 49.33 (0.4) | 29.52 (0.3) |
>
> 2. We would also like to clarify that the appendix already includes a noisy-target robustness study. In **Appendix J.2**, we corrupt a fraction of target validation examples by swapping answer prompts across examples and measure GSM8K performance as the noise ratio increases. TRIM degrades gradually rather than catastrophically, indicating that the aggregate token fingerprints remain reasonably stable even under substantial target noise. We will make this result more prominent in the revision.
> 3. We thank the reviewer for this important point. In principle, TRIM should extend to long-context instruction tuning as well.
>    - The main practical challenge is that efficient long-context training typically uses FlashAttention-style kernels. TRIM fingerprinting requires token-level attention statistics for saliency computation, which are not directly exposed in this setting. To address this, we implemented a preliminary workaround that computes the required saliency statistics through a chunked Q/K reconstruction procedure on top of token hidden states, rather than materializing the full attention matrix at once. This makes the experiment feasible, but introduces additional latency. Developing saliency estimators that are more naturally compatible with FlashAttention remains future work.
>    - Using this implementation, we add a preliminary long-context experiment based on **LongLoRA/LongAlpaca-12k** [5]. We split LongAlpaca-12k into train/validation/test subsets using stratified length bins so that each split preserves the long/short composition of the dataset. This mirrors our in-domain selection setup in Sec. 4.5. We use a LLaMA-2-7B backbone with a LongLoRA-style recipe, keeping the base architecture unchanged. As a simple first measure, we report held-out assistant-token perplexity. *Full-data fine-tuning achieves perplexity 3.19*. In the much harder 50% subset regime, *random selection gives perplexity 4.24*, while **TRIM improves this to 3.67**. This suggests that TRIM can be extended to long-context settings.
> 4. We also clarify that the appendix already includes qualitative fingerprint analysis. **Appendix L** presents representative fingerprinted tokens together with Nearest-Fingerprinted Fallback (NFF) examples, showing that the fingerprints capture task-relevant lexical and formatting patterns. In addition, our GSM8K qualitative case study shows that, under the answer-side TRIM configuration, high saliency is assigned to intermediate arithmetic derivations and answer-format cues (e.g., explicit computation steps and final numeric resolution), and that high-scoring selected candidates exhibit the same multi-step solution structure. Taken together, these results illustrate both which tokens/spans TRIM identifies as salient and how those fingerprints influence the selection process.
>
> References
> ---
> [1] Liu, Zhu, et al. "Fantastic semantics and where to find them: Investigating which layers of generative LLMs reflect lexical semantics." 2024
>
> [2] Gao, Chongyang, et al. "Higher layers need more LoRA experts." 2024
>
> [3] Men, Xin, et al. "Shortgpt: Layers in large language models are more redundant than you expect." 2025
>
> [4] Askari, Hadi, et al. "LayerIF: Estimating Layer Quality for Large Language Models using Influence Functions." 2025
>
> [5] Chen, Yukang, et al. "Longlora: Efficient fine-tuning of long-context large language models." 2023

---

> > ### Author Rebuttal · Reviewer_iyQr · 2026-04-02
> >
> > Thanks for the response. My concerns have been fully addressed, I will keep my rating as 4.

---

> > > ### Author Response · Authors · 2026-04-04
> > >
> > > Thank you again for the thoughtful review and for taking the time to read our rebuttal. We are very glad that the additional layer-wise ablation, noisy-target robustness study, long-context evaluation, and qualitative fingerprint analysis addressed your concerns.
> > >
> > > Since these were the main issues raised in the review and now appear resolved, we would appreciate it if you could consider increasing your score. We sincerely appreciate your consideration.

---

### Official Review · Reviewer_eF8k · 2026-03-11

**Soundness:** 3
**Presentation:** 3
**Significance:** 3
**Originality:** 3
**Overall Recommendation:** 4
**Confidence:** 4

**Summary:**

The paper addresses the computational cost and biases associated with selecting instruction-tuning coresets for large language models. Existing approaches typically rely on sample-level gradient signals, which are computationally expensive and can exhibit length bias. To address this, the authors propose TRIM (Token Relevance via Interpretable Multi-layer Attention), a forward-only framework that scores data at the token level. TRIM uses the model's multi-layer attention to calculate token saliency and constructs representational "fingerprints" from a small target set of examples. It then ranks candidate data by computing the cosine similarity between the candidates' token hidden states and these target fingerprints. The authors evaluate TRIM across multiple reasoning benchmarks, reporting that it achieves comparable or higher accuracy than baseline methods at a lower computational cost, while selecting a broader distribution of sequence lengths.

**Compliance With Llm Reviewing Policy:**

Affirmed.

**Key Questions For Authors:**

Questions

1. The method assumes attention-derived saliency proxies task relevance, but lacks formal theoretical bounds linking it to the loss. Did you evaluate whether TRIM's saliency scores (or final sample scores) formally correlate with gradient-based influence metrics or loss-based importance scores?
2. The saliency score combines row and column attention using a fixed equal split (0.5 each). Was this empirically tuned, and did you test if dynamically learning or tuning these weights during the warmup phase improves results for different downstream tasks?
3. How frequently is the Nearest-Fingerprinted Fallback (NFF) mechanism invoked in practice during candidate scoring? Furthermore, while mapping to static embeddings may work for general NLP, how does this fallback handle highly structured domains (like code or math) where unseen tokens or domain-specific operators might not map accurately using simple semantic proximity?

**Limitations:**

Partially.

The Impact Statement discusses potential risks such as amplification of biased or harmful content during data selection. However, the paper could more explicitly discuss methodological limitations of the approach.

**Strengths And Weaknesses:**

Strengths

1. The method is evaluated across several diverse benchmarks, including MMLU, BIG-Bench Hard, and TyDiQA. The approach appears capable of selecting task-relevant data (e.g., math reasoning examples for GSM8K) from a general candidate pool.
2. The method is computationally efficient. It runs substantially faster than gradient-based selection approaches such as LESS while achieving comparable performance.
3. The approach mitigates the length bias commonly observed in sample-level selection methods, enabling the selection of longer reasoning-heavy examples.
4. The paper demonstrates strong transferability: subsets selected using smaller models can be used effectively to fine-tune larger models such as LLaMA-2-13B or models with different architectures such as Mistral-7B.
5. The paper includes useful ablation studies that help justify several design choices, including the use of IDF weighting and fallback penalties.
6. The work introduces a practical shift from sample-level gradient scoring to token-level scoring based on attention-derived signals.
7. The paper is clearly written and well organized. The problem setup and the two-stage pipeline are easy to follow.

Weaknesses

1. The method relies on an intuitive heuristic rather than a formal theoretical justification. The approach assumes that attention-derived saliency correlates with task relevance, but the paper does not provide evidence that this signal aligns with gradient-based importance or the loss function.
2. The fallback mechanism for unknown tokens relies on static embeddings, which may introduce noise when evaluating highly contextualized sequences.
3. Some key hyperparameters lack justification. In particular, the equal weighting of row and column attention signals appears heuristic, and it is unclear whether alternative weighting strategies were explored.
4. The paper would benefit from qualitative examples illustrating which tokens are identified as salient and how these influence the selection process.
5. The claim that the method is “forward-only” is somewhat overstated, since the pipeline still requires a short warmup fine-tuning phase.
6. The underlying mathematical components (e.g., TF-IDF weighting, cosine similarity, entropy) are standard; the primary contribution lies in combining these elements into a practical pipeline rather than introducing fundamentally new algorithmic ideas.

---

> ### Author Rebuttal · Authors · 2026-03-30
>
> We thank the reviewer for providing invaluable feedback and helping us strengthen the paper. We provide clarifications to address the points raised in the weaknesses and questions below.
> 1. We agree that a formal theoretical bridge between TRIM’s saliency signal and influence-function-based importance would be valuable.
>    - That said, we do not view classical influence functions as a universal gold standard for modern LLM finetuning. Influence functions are local first-order approximations around a trained optimum, and prior work has shown that they can be fragile in deep, non-convex models [1, 2]. Recent work has also shown that training dynamics differ in finetuning [3], indicating that the usefulness of previous influence metrics may depend on evolving token-level behavior rather than a single static sample-level score. For these reasons, we believe that a more faithful notion of utility for instruction tuning would ideally be token-aware, but this is expensive. TRIM therefore uses an empirically motivated token-level proxy for coreset selection, and we do not claim it is an influence function.
>    - Following the reviewer’s suggestion, we calculated the Spearman rank correlation between TRIM and LESS sample scores over the candidate pool and observed scores of  0.97 on MMLU, 0.69 on TyDiQA, 0.68 on BBH, and 0.32 on GSM8K. We interpret this as follows: TRIM aligns closely with LESS on knowledge-heavy tasks such as MMLU, while the lower correlation on GSM8K suggests that the two methods capture different utility signals on reasoning-heavy problems (A more detailed explanation of the samples chosen is provided in **Appendix K**).
> 2. The row and column saliency were given equal weightage, as conceptually, the two terms capture complementary views of token importance: one reflects how a token distributes attention, while the other reflects how much it is attended to by the rest of the sequence. We also ablated over the different weightage values and provided the results in Table R3 below.
>
> **Table R3: Performance of Llama 2 7 B fine-tuned  on 5% coresets targeted on MMLU**
>
> | $(w_Q, w_K)$ | (0, 1) | (0.1, 0.9) | (0.25, 0.75) | (0.5, 0.5) | (0.75, 0.25) | (0.9, 0.1) | (1, 0) |
> | ---------- | ------ | ---------- | ------------ | ---------- | ------------ | ---------- | ------ |
> | Mean Accuracy (%)       | 47.24  | 47.83      | 49.32        | 49.33      | 49.21        | 48.34      | 47.43  |
>
> 3. NFF is invoked whenever a candidate token lacks a corresponding entry in the target fingerprint set; it is not an out-of-vocabulary mechanism, since all experiments use a shared tokenizer. Because TRIM builds a compact target-conditioned fingerprint inventory from a small target validation set, such unmatched tokens can arise in practice on diverse source pools. We therefore view NFF as a pragmatic coverage mechanism that extends token-level matching beyond the directly fingerprinted set, rather than as a separate source of supervision. We also agree that static embedding proximity is an imperfect approximation in highly structured domains such as code or formal math, where operators and symbols may have brittle semantics. For this reason, we do not treat NFF as equivalent to a direct fingerprint match: fallback matches are explicitly downweighed by the NFF penalty. The current submission already includes qualitative NFF examples in **Appendix L** (Table 18), where some mappings are intuitive lexical variants (e.g., “may” to “might”), illustrating why the mechanism is useful as a simple backoff, while also underscoring that it is not intended to be a perfect semantic resolver. Empirically, despite this approximation, TRIM remains effective in reasoning-heavy settings such as GSM8K, suggesting that the fallback is practically useful even when exact fingerprint overlap is incomplete.
> 4. We also agree that qualitative analysis improves interpretability. To address this, we will include a GSM8K case study showing which tokens receive high saliency scores and how these affect the selected candidate examples. For example, in one representative target example, the model assigns high saliency to the solution steps computing the total pages and per-book average (e.g., `80 × 12 = 960` and `960 / 6 = 160`).  A high-scoring selected candidate from the final coreset exhibits the same kind of multi-step numeric reasoning structure: it contains explicit intermediate computations such as `100 / 2 = 50`, `50 × 20 = 1000`. Qualitatively, this illustrates the intended behavior of TRIM, selecting candidate examples whose responses contain similar patterns as the target task, biasing the coreset towards examples with similar reasoning patterns.
>
> References
> ---
> [1] Basu et al., Influence Functions in Deep Learning Are Fragile (2021).
>
> [2] Epifano et al., Revisiting the Fragility of Influence Functions (2023).
>
> [3] Ren & Sutherland, Learning Dynamics of LLM Finetuning (2025)

---

> > ### Author Rebuttal · Reviewer_eF8k · 2026-04-03
> >
> > I thank the authors for their thorough rebuttal, which successfully resolved my primary technical concerns. I am keeping my score of Weak Accept, as I feel this rating is sufficiently high and accurately reflects the paper's contributions.

---

> > > ### Author Response · Authors · 2026-04-04
> > >
> > > Thank you again for taking the time to read our rebuttal. We are glad that we were able to address the concerns raised in the weaknesses and questions around saliency correlation, weighting ablations, NFF behavior, and qualitative interpretability.
> > > Given that these concerns now seem resolved, we would appreciate it if you could consider updating the score to reflect this positive assessment. We thank you again for your time and comments.

---

### Official Review · Reviewer_k7uZ · 2026-03-13

**Soundness:** 3
**Presentation:** 3
**Significance:** 3
**Originality:** 3
**Overall Recommendation:** 4
**Confidence:** 3

**Summary:**

TRIM proposes a token-level data selection method for efficient targeted instruction tuning. Specifically, it computes attention-based saliency on target samples to construct token-level fingerprints, and then scores and select candidate training samples based on their similarity to these fingerprints. Since the method relies only on forward passes without gradient computation, it is computationally efficient compared to gradient-based approaches.

**Compliance With Llm Reviewing Policy:**

Affirmed.

**Final Justification:**

Having considered all the authors’ responses, I still believe that 4 (Weak Accept) is the appropriate overall recommendation.

**Key Questions For Authors:**

- The paper provides a helpful analysis of length bias. However, I found the evidence for the loss-centric bias discussed in the introduction less direct. It would strengthen the paper if the authors could clarify whether there is direct empirical support for this claim, or provide additional analysis.

- I am concerned that TRIM aggregates all occurrences of a token class into a single fingerprint across the whole target set. For heterogeneous benchmarks such as BBH, where different validation examples come from different subtasks, this may blur task-specific meanings of the same token and lead to information loss.

- IDF reweighting is introduced in the main methodology, but from the appendix it seems that it may not actually be used in the experiments.

**Limitations:**

yes

**Strengths And Weaknesses:**

- The paper is well written, and the motivation is clear. The authors highlight the limitations of existing approaches, particularly the computational cost and issues such as length bias and loss-centric bias. They address these issues through a token-level approximation. The idea of using attention signals to identify important tokens for data selection is also interesting and novel. The experiments show that the proposed method performs better than previous approaches. It also improves efficiency in terms of computational cost (runtime) compared to prior methods.

- However, the proxy of constructing token-class fingerprints using attention and selecting similar samples appears less intuitive compared to existing gradient- or Hessian-based approximation methods. Although the results show improvements, the performance gains over prior methods do not appear to be very large (Table 1).

- In addition, comparing the proposed method with more recent approaches (e.g., Task-Specific Data Selection for Instruction Tuning via Monosemantic Neuronal Activations) would further strengthen its advantages.

---

> ### Author Rebuttal · Authors · 2026-03-30
>
> We thank the reviewer for providing their insights and helping us improve the paper. We address the weaknesses and questions raised during the review period below, and hope it provides the necessary clarifications.
>
> - **W1**: We appreciate this point. While the average gains in **Table 1** are sometimes modest, they are achieved with a substantially more efficient selection pipeline (**Section 5**). Moreover, the improvements are more pronounced on certain downstream tasks, most notably GSM8K, where TRIM improves over LESS by **nearly 9% in Table 2**, while remaining competitive on the others. We will clarify this more explicitly in the revised version. We agree that the intuition behind TRIM should be stated more clearly. Classical influence-based methods rely on mathematically derived sample-importance metrics, typically based on gradient- or curvature-related approximations. TRIM instead is designed to match target-relevant patterns at the token level from the model’s own representation space: it uses hidden states to represent token context, and saliency scores to estimate which target tokens are most important in context. Candidate samples are then preferred when their token representations align well with these salient target patterns. Thus, TRIM is not intended as an influence-function approximation, but as a token-aware retrieval signal for instruction-tuning utility. We will make this intuition clearer in the revision, including an illustrative example.
> - **W2**: We agree that comparison to MONA [1] would further strengthen the empirical positioning; however, to the best of our knowledge, we were unable to identify an official public implementation at the time of writing, and we did not want to include a potentially unfair reimplementation-based comparison. We will add a discussion of this method in the revision. Conceptually, MONA and TRIM are quite different: although MONA begins from token activations, it ultimately aggregates token-level activations into a single sample-level embedding and then compares each source sample to an averaged task prototype in that sample space. In contrast, TRIM is token-level and directly target-conditioned, constructing token-class fingerprints from the target validation set and selecting examples based on fine-grained token alignment rather than sample-level similarity.
> - **Q1**: We thank the reviewer for the helpful observation on the loss bias. We agree that, in the current draft, the evidence for the “loss-centric bias” claim is more indirect than the evidence we provide for length bias, and we will clarify this in revision. More precisely, we claim that sample-level loss/gradient proxies may obscure token-level importance by reducing each example to a single sequence-wise score. This can disadvantage examples where only a few tokens are strongly relevant to the target task, even if the overall sequence loss is not extreme. This intuition motivates TRIM’s token-level scoring and is supported indirectly by our ablations in Appendix H, showing that preserving both average and peak token-level signals is more effective than relying on a purely averaged signal. We will revise the wording accordingly so that this point is presented as a motivation grounded in our design, rather than as a separate, fully established empirical claim.
> - **Q2**: For the concern about aggregating all occurrences of a token class into a single fingerprint across a heterogeneous target set such as BBH, we agree that this is a modeling simplification and that finer-grained subtask-conditioned fingerprints could be an interesting future extension. However, in TRIM, this aggregation is used only for data selection, not to modify the downstream learning objective or training dynamics. Moreover, the fingerprint is formed from contextual token representations/attention-derived statistics, so it does not collapse tokens purely at the lexical level; rather, it summarizes the average target-conditioned behavior of that token class across the validation set. In heterogeneous benchmarks like BBH, this average target signal is in fact what we want the selector to capture: it biases selection toward examples whose tokens are broadly aligned with the overall target distribution, while the full training examples are still used unchanged during fine-tuning. Empirically, this approximation appears sufficient, as evidenced by the downstream results, including BBH.
> - **Q3**: Regarding IDF reweighting, it is used in the experiments; we apologize if this was not sufficiently clear in the main text. We provide both the implementation details and ablations demonstrating its effect in **Appendix C**, and we will make this explicit in the revision.
>
> References
> ---
> [1]  Ma et. al., Task-Specific Data Selection for Instruction Tuning via Monosemantic Neuronal Activations, 2025.

---

> > ### Author Rebuttal · Reviewer_k7uZ · 2026-04-03
> >
> > Thank you for the response. I think it would be helpful to include this discussion in the paper. I will keep my score unchanged.

---

> > > ### Author Response · Authors · 2026-04-04
> > >
> > > Thank you for taking the time to read our rebuttal and follow-up. We are glad the additional discussion was helpful, and we agree that these clarifications strengthen the paper.
> > >
> > > To summarize, in the revision, we will make the intuition behind TRIM more explicit, clarify the scope of the loss-centric bias claim, state more clearly that IDF reweighting is used in the experiments, and better explain why the target-conditioned aggregation is used only for data selection and does not alter the downstream training objective as discussed.
> > >
> > > Since the concerns now seem addressed, we would appreciate it if you could consider increasing your score. We sincerely appreciate your thoughtful review and thank you again for your time and comments.

---

### Official Review · Reviewer_jzPS · 2026-03-18

**Soundness:** 2
**Presentation:** 3
**Significance:** 2
**Originality:** 3
**Overall Recommendation:** 3
**Confidence:** 3

**Summary:**

The goal of Data-Efficient Instruction Tuning is to automatically select a small yet high-quality representative subset from a large pool of candidate instruction data, so as to achieve high-quality fine-tuning at lower cost. The authors point out that existing coreset selection methods rely on sample-level gradient signals, are computationally expensive, and ignore token-level structural information. To address this, they propose TRIM. The method first uses a small number of target samples to extract token fingerprints weighted by attention-based saliency, then matches these fingerprints with contextual token representations in the candidate samples, and finally aggregates the matching results into sample-level scores for selection. Experiments show that TRIM can match or outperform some methods on multiple tasks, while also incurring significantly lower selection cost. Based on these findings, the authors argue that token-level structure-aware data selection is more suitable than traditional sample-level methods for efficient instruction tuning.

**Compliance With Llm Reviewing Policy:**

Affirmed.

**Key Questions For Authors:**

#### Key Questions
1. The paper describes row saliency (query entropy) and column saliency (key attention quality) as complementary signals and combines them with equal weights, but it does not provide ablations for \(w_Q = 1.0, w_K = 0\) and \(w_Q = 0, w_K = 1.0\). If all token weights are replaced with uniform weights (\(\alpha_{v,i} = 1\), i.e., without using attention-derived saliency), does TRIM still maintain an advantage over RDS, which also uses forward representations but at the sample-similarity level?
2. For different task settings, how should the hyperparameters for mean, max, and coverage be selected? Are the hyperparameters consistent across the different experimental settings, and could the authors provide the corresponding results?
3. Could the authors explain more clearly the relationship between this paper and QCS (Chen et al., 2025b)? Since both adopt token-level designs, what are the specific advantages of this work?

**Limitations:**

Yes

**Strengths And Weaknesses:**

#### Strengths
1. The core method of this paper is very interesting and innovative. Unlike prior coreset selection methods that rely heavily on sentence-level features, the authors propose a token-level approach that computes token fingerprints from prototype data based on attention and hidden states, using them to select high-value data. I personally believe that focusing on common important words and expression patterns is a highly valuable approach to data selection.
2. The authors report variance in their experimental results, which reflects a very rigorous attitude toward reporting results. I believe this is commendable.
3. The authors provide an estimate of the time cost of the data selection algorithm, clearly demonstrating the superiority of their method in both performance and computational overhead.

#### Weaknesses
1. The existence of the QCS paper reduces the value of this work, especially since the authors exclude it from the experiments on the grounds of “excessive computational cost” without providing any quantitative analysis. Since both methods operate at the token level, the authors should more clearly articulate the differences between them. They should consider providing:
   1. an estimated runtime of QCS on the same 270k candidate pool with a single H200 GPU, even if based only on the complexity reported in the QCS paper;
   2. reference accuracy numbers of QCS on MMLU/GSM8K, either from the original paper or from public reproductions.
2. The proposed method appears somewhat heuristic and dependent on empirical hyperparameter tuning. For example, in Section 3.2, the sample score is defined as $ S(c) = w_\mu \cdot \text{mean} + w_m \cdot \text{max} $, but Appendix C additionally includes a coverage term, while the pseudocode in Algorithm 2 is consistent with Eq. 13 and does not include coverage. Given that the ablation study in Table 10 shows that removing the coverage term causes MMLU to drop by 0.68, and that the default configuration in Appendix H includes the coverage term, the actual experiments seem to use Eq. 15 rather than Eq. 13.
3. The paper lacks sufficient rigor in its presentation.
   1. From the main tables, the proposed method does not show a clear improvement over Less, so the SOTA claims in the abstract and introduction appear overstated.
   2. In Section 3.1, the claim that “the two signals are complementary” lacks direct support. The paper does not include ablation experiments that substantiate this conclusion. The authors should add ablations under three settings, {wQ = 0, wQ = 1, wQ = 0.5}, as well as a uniform-weight baseline.

---

> ### Author Rebuttal · Authors · 2026-03-30
>
> We thank the reviewer for providing invaluable feedback and providing us the opportunity to strengthen our work. We provide some clarification to your questions and points raised in the weakness section below.
> ### W1, Q3. Relationship to QCS
> - While both TRIM and QCS are token-aware methods, they differ substantially in how they operate. TRIM requires a brief warmup, and then constructs target fingerprints and scores each candidate example before ranking and selecting the top examples. QCS formulates the selection as a bi-level (sequence and token levels) optimization problem. The targeted-IT variant proposed in the paper first obtains the selection variables on the candidate pool, then trains separate transfer selector modules on the source data, and then adapts those modules on the target validation set.  In this sense, it is a direct target-conditioned scorer, while QCS is an indirect transfer-based selector.
> - This difference in mechanism leads to the difference in computational overheads. Using the notation in our paper, we model the selection-stage cost of TRIM (warmup + scoring) as $O(3f\gamma NT + fN)$ in Section 5. Similarly, we can estimate the targeted-QCS selection pipeline as $O(3fNT_\text{solver} + 3c_h fN T_\text{transfer} + 3c_h fQ T_\text{adapt} + fN)$, where $c_h=2$ accounts for the separate sequence- and token-level transfer selectors in Algorithm 2 of the paper. This reflects: (i) solving QCS on the source, (ii) training transfer selector heads on the source, (iii) test-time adaptation on the target, and (iv) a final forward sweep for selection. Hence, with $c_h=2$ and $Q \ll N$, and using $4$ epochs for both (i) and (ii) as stated in the paper, a concrete estimate of the cost is approximately $O(37fN)$. By contrast, TRIM’s selection cost is  $O(1.6fN)$ for $\gamma =0.05$ and $T=4$. Therefore, while both are linear in pool size at a high level, QCS remains training-dominated throughout selection, whereas TRIM is forward-only after a brief warmup. We will add a detailed analysis in the revision.
> - We did not include QCS as an experimental baseline because we could not find an official public implementation at submission time. Without a faithful reimplementation, we felt that a direct numerical comparison would be potentially misleading. In particular, comparing absolute scores across papers is difficult because the full-data baselines differ, suggesting that the underlying experimental conditions are not identical. To still provide context, following the reviewer’s suggestion, we provide Table R1, a normalized comparison of each method’s 5% coreset relative to the corresponding full fine-tuning result reported in its own paper. We will revise the paper to make these points clear.
>
> **Table R1 5% coreset performance on Llama 2 7B. Smaller drops are better; + means outperforming full-data finetuning**
>
> | Method                  | MMLU   | BBH   | TydiQA | GSM8K |
> | ----------------------- | ------ | ----- | ------ | ----- |
> | TRIM vs Full Finetuning | -0.79  | -3.11 | +2.62  | -0.73 |
> | QCS vs Full Finetuning  | -1.60  | -0.80 | +2.60  | -5.20 |
> ### W2. Typographical Errors
> We appreciate the reviewer for pointing out a typographical error. To clarify, we do use coverage, and eq (13) and the pseudocode must include it. We will correct these errors.
> ### W3.1. Positioning with SOTA
> We agree that the claims can be phrased more carefully, and we will revise the abstract/introduction to avoid overstating uniform superiority over LESS. Our main point is that TRIM provides a much more efficient selection procedure while still delivering competitive or better downstream performance. Although TRIM does not dominate LESS on every benchmark, it does outperform LESS clearly on some important settings, e.g., GSM8K by nearly 9 points, and remains competitive on others.
> ### W3.2., Q1, Q2. Hyperparameters
> - We conducted a fine-grained ablation varying the relative weights of row and column saliency as suggested,  and provided the results in Table R3 (please see the comment to reviewer eF8k).
> - We used the same hyperparameters across all task settings. We set the mean and max weights equally, and selected the coverage coefficient once via a constrained grid search. Empirically, this setting performed the best, as shown in the appendix.
> - We also show results of the performance of TRIM without saliency weighting (equal token weightage) in Table R2. We find that this method of scoring candidate samples is not very effective.
>
> **Table R2: Performance of Llama 2 7B finetuned on 5% coresets**
>
> | Method                | MMLU       | TydiQA      | BBH         | GSM8k       |
> | --------------------- | ---------- | ----------- | ----------- | ----------- |
> | TRIM                  | 49.33(0.4) | 56.62(0.1)  | 39.73(0.8)  | 29.52(0.3)  |
> | TRIM without saliency (equal weightage) | 46.12(0.3) | 46.34 (0.1) | 36.12 (0.2) | 18.35 (0.8) |
> | RDS                   | 45.27(0.6) | 46.10(0.2)  | 35.13(0.5)  | 17.45(0.2)  |

---

> > ### Author Rebuttal · Reviewer_jzPS · 2026-04-08
> >
> > The authors acknowledged my point, and I believe this score is reasonable.

---

> > > ### Author Response · Authors · 2026-04-08
> > >
> > > Thank you for taking the time to read our rebuttal and for indicating that your concerns have been fully resolved. We sincerely appreciate that.
> > >
> > > Given that the original concerns now appear to be addressed, we would appreciate it if you could consider increasing the score to reflect your updated assessment better. Thank you again for your time and consideration.

---

### Decision · Program_Chairs · 2026-04-30

**Decision:**

Accept (regular)

**Comment:**

This paper proposes TRIM as a forward-only, efficient, instruction fine-tuning data selection method based on token-level pattern matching. In particular, they extract the attention-based "fingerprints" for tokens in a small target set of prototypes and then match these fingerprints with contextual token representations in the candidate samples (by their cosine similarity), and finally aggregate the matching results into sample-level scores for selection. The authors evaluate TRIM across multiple reasoning benchmarks, reporting that it achieves comparable or higher accuracy than baseline methods at a lower computational cost, while selecting a broader distribution of sequence lengths.

Reviewers found the idea innovative, and it overcomes the computational bottleneck of gradient-based selection. That being said, the token-level footprint matching is less intuitive than gradient or Hessian-based strategies. The sample score finally used is unclear and needs more clarification. More ablation studies of the sample score function, the layers to extract footprints, and a few hyperparameters are lacking. And the paper should be more careful when claiming advantages over existing baselines. That being said, the reviewers also found that the empirical results strongly support the effectiveness of the proposed method. The rebuttal fully resolved three out of four reviewers' concerns. Based on the final ratings and justification of the reviewers, I suggest a weak acceptance of this work.